# Engineering a local acid-like environment in alkaline medium for efficient hydrogen evolution reaction

Hao Tan[1,2], Bing Tang[1,2], Ying Lu[1,2], Qianqian Ji[1], Liyang Lv[1], Hengli Duan[1], Na Li[1], Yao Wang[1], Sihua Feng[1], Zhi Li[1], Chao Wang [1✉], Fengchun Hu[1], Zhihu Sun [1✉] & Wensheng Yan [1✉]

Tuning the local reaction environment is an important and challenging issue for determining electrochemical performances. Herein, we propose a strategy of intentionally engineering the local reaction environment to yield highly active catalysts. Taking $Pt^{\delta-}$ nanoparticles supported on oxygen vacancy enriched MgO nanosheets as a prototypical example, we have successfully created a local acid-like environment in the alkaline medium and achieve excellent hydrogen evolution reaction performances. The local acid-like environment is evidenced by operando Raman, synchrotron radiation infrared and X-ray absorption spectroscopy that observes a key $H_3O^+$ intermediate emergence on the surface of MgO and accumulation around $Pt^{\delta-}$ sites during electrocatalysis. Further analysis confirms that the critical factors of the forming the local acid-like environment include: the oxygen vacancy enriched MgO facilitates $H_2O$ dissociation to generate $H_3O^+$ species; the F centers of MgO transfers its unpaired electrons to Pt, leading to the formation of electron-enriched $Pt^{\delta-}$ species; positively charged $H_3O^+$ migrates to negatively charged $Pt^{\delta-}$ and accumulates around $Pt^{\delta-}$ nanoparticles due to the electrostatic attraction, thus creating a local acidic environment in the alkaline medium.

---

[1] National Synchrotron Radiation Laboratory, University of Science and Technology of China, Hefei 230029, P. R. China. [2]These authors contributed equally: Hao Tan, Bing Tang, Ying Lu. ✉email: chaowng@ustc.edu.cn; zhsun@ustc.edu.cn; ywsh2000@ustc.edu.cn

The development of a highly-efficient and low-cost pathway to produce renewable energy resources has been one of the top priorities in the science community in recent decades. Electrochemistry approaches play essential roles in this field due to the low reaction activation energy and high reaction rate and energy efficiency[1,2]. The kinetics of electrode reactions depend strongly on both the nature of electrode materials and the local concentrations of solution constituents in the vicinity of the catalytic sites (hereafter, called the local reaction environment)[3]. Thus far, the efficiency of the electrode reactions is mainly increased by ameliorating the catalyst materials through various methods, such as controlling the crystal facet[4], doping heteroatoms[5], creating dual active sites[6], engineering defects[7], and strains[8], and so on. In most cases, these conventional strategies can only tailor their atomic structures, electronic states, and thereby catalytic properties in a gradual or mild way. To seek breakthroughs in electrode reactions, one needs to recall the elementary steps in electrochemistry. All the electrochemical processes involve the non-covalent interactions between the atoms and molecules of the electrocatalyst surface and the reaction intermediates generated by the solvent or solute molecules, leading to the formation of intermediate complexes or nonuniform distribution of the ionic species. Therefore, the local reaction environment around catalysts also plays a vital role in electrode processes[9–11]. Tuning the local reaction environment through multiple physicochemical effects between the substrate, metal, and reaction intermediate thus provides an alternative way to promote the electrocatalytic performance and guide the higher efficiency electrocatalyst design. This task is challenged by the lack of facile and practical strategies to engineer the local reaction environment, as well as by the difficulties in identifying the weak signals arising from interfacial structures under complex reaction conditions[12].

The hydrogen evolution reaction (HER) is one of the most classic electrochemical reactions in both academic research and industry applications[13–15]. Industrial plants prefer to use water-based alkaline solution instead of acidic solution as hydrogen sources; the reason is that the alkaline HER can both avoid using the high-cost proton exchange membrane and alleviate the problem of slow electron-transfer kinetics of the oxygen evolution reaction (OER) on the anode within an electrolyzer under acidic conditions[16,17]. However, the alkaline HER also has its drawbacks. The principal one is the low conversion efficiency. Taking Pt-based catalysts, which are regarded as the most efficient HER catalysts as an example, the HER conversion efficiency in alkaline condition is two or three orders of magnitude lower than that in acidic solutions[18]. The alkaline HER kinetics are still elusive, and several schools of thought on the slow kinetics of the alkaline HER have been proposed, including the hydrogen binding energy (HBE) theory[19], water-dissociation theory[20–22], and interface water and/or anion transfer theory[23,24]. Nevertheless, the general consensus is that during the alkaline HER, the sluggish Volmer step directly or indirectly impacts the rate-determining step, but this step is unnecessary in an acidic solution. One can expect that if we could create a local acid-like environment for the HER in an alkaline medium where the corresponding OER occurs, the adverse factors of the alkaline HER and acidic OER will be solved fundamentally, and both the HER and OER could proceed at a high rate. The feasibility of this strategy is illuminated by recent studies showing that the interfacial microstructure of the electrolyte can be modified by the physicochemical properties of the solid surface[9]. Therefore, modulating the local reaction environment by selecting a suitable system to create a local acid-like environment, will provide a route to design highly-efficient HER catalysts. Previous studies have revealed that MgO, $Al_2O_3$ and $Ni(OH)_2$ surfaces facilitate the dissociation of $H_2O$ molecules,

giving rise to $H^+$ groups[25–28]. However, these free protons cannot accumulate spontaneously around the catalytic sites to form a concentrated acidic region. Thus, an electrocatalyst system capable of driving the aggregation of $H^+$ groups should be developed.

In this work, we propose a practical pathway for engineering a local acid-like reaction environment to design highly-efficient alkaline HER catalysts. By virtue of multiple physicochemical interactions between the substrate, metal active site, and reaction intermediate, we selected Pt/MgO as the prototypical example to construct an acid-like reaction environment in an alkaline medium. Operando Raman spectroscopy, synchrotron radiation Fourier transformed infrared spectroscopy (SR-FTIR) spectroscopy, and X-ray absorption near-edge spectroscopy (XANES) confirm the generation of massive amounts of $H_3O^+$ intermediates on the MgO surface and accumulation around negatively-charged Pt ($Pt^{\delta-}$). The local acid-like reaction environment leads to an extraordinary HER performance, with a very low overpotential of 39 mV at 10 mA cm$^{-2}$, which is much better than the value of 62 mV for 20 wt% Pt/C and close to the acidic HER behavior of 20 wt% Pt/C (33 mV). This system also has tenfold higher mass activity than that of the 20 wt% Pt/C electrodes in an alkaline medium and 2.5-fold higher than that of 20 wt% Pt/C in acidic medium at −39 mV vs. RHE. Experimental characterizations and first-principles calculations suggest that the oxygen vacancy-rich MgO is favorable for water dissociation, and the electronic interaction between the MgO and Pt nanoparticles drives electron transfer from $V_O$-MgO to Pt, giving rise to the formation of negatively-charged $Pt^{\delta-}$ species. Then the $Pt^{\delta-}$ accelerates $H_3O^+$ migration and an acid-like environment is formed around the $Pt^{\delta-}$ in an alkaline medium, thus boosting the HER in this alkaline medium. We believe that this finding will contribute to future explorations in other important solution-dependent fields, such as surface, energy and environmental science.

## Results

**Structural and morphological characterizations.** To synthesize Pt/MgO catalysts with strong electronic metal-support interactions (EMSIs) and an abundance of oxygen vacancies, we chose MgMOFs as the precursor for the MgO support (Supplementary Fig. 1), and immersed Pt ions into the pores of MgMOFs, followed by annealing at 700 °C in an atmosphere of nitrogen and oxygen. As revealed by the thermogravimetric analysis shown in Supplementary Fig. 2, the formation of Pt nanoparticles on MgO consists of three steps. The successful synthesis of Pt/MgO is confirmed by powder X-ray diffraction (XRD) as shown in Fig. 1a. The diffraction peaks at 2θ angles of 39.8° and 46.4° are indexed to the (111) and (200) planes of fcc Pt nanocrystals, respectively, and the other diffraction peaks correspond to the MgO support. Figure 1b presents a transmission electron microscopy (TEM) image of Pt/MgO, indicating the uniform dispersion of Pt nanoparticles on the MgO nanosheet support (the MgO nanosheet is shown in the inset and Supplementary Fig. 3). The sizes of the Pt nanoparticles are ~5 nm (Supplementary Fig. 4). From the high-resolution TEM (HRTEM) image in Fig. 1c, the lattice fringes of 0.196 and 0.209 nm are ascribed to the (200) planes of fcc Pt and (200) plane of MgO, respectively. Energy-dispersive X-ray spectroscopy (EDS) mapping qualitatively reveals that the Pt nanoparticles are homogeneously dispersed on the MgO support (Fig. 1d). From the atomic force microscopy (AFM) images as shown in Supplementary Fig. 4, a majority of the Pt/MgO nanosheets are approximately 200–500 nm in width and ~4 nm in thickness. The EXAFS fitting results (Supplementary Fig. 6 and Table 1) show that the coordination number of the first shell of Pt is 10.1, which is

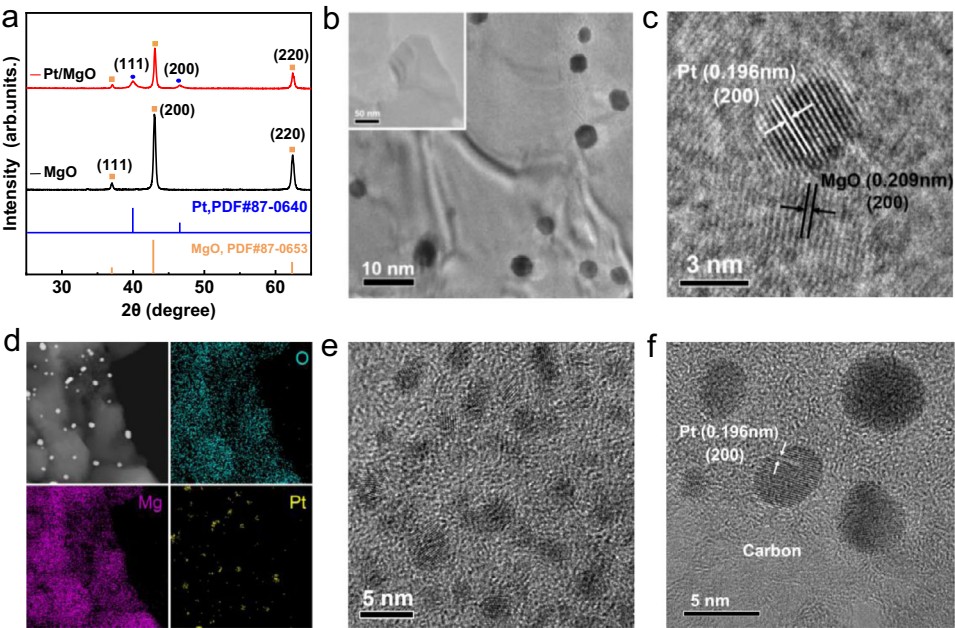

**Fig. 1 Structural characterizations. a** XRD patterns of Pt/MgO and MgO. **b** TEM image of Pt/MgO, the inset shows the MgO nanosheet. **c** HRTEM image of Pt/MgO. **d** Elemental mapping images of Pt/MgO. **e** TEM and **f** HRTEM images of Pt/C.

significantly lower than the value (12) of Pt foil. The decreased coordination number is mainly due to the existence of a significant quantity of unsaturated coordination bonds on the surface of nanosized materials. Inductively coupled plasma atomic emission spectroscopy (ICP–AES) analysis reveals that the content of Pt in the Pt/MgO samples is 3.2 wt%. Taken together, we have demonstrated the successful synthesis of Pt/MgO nanosheets by the two-step hydrothermal-annealing method. For comparison, the TEM and HRTEM images of commercial 20 wt% Pt/C are shown in Fig. 1e, f, which exhibit a similar size but a much higher density of nanoparticles (Supplementary Fig. 7).

**Confirmation and formation mechanism of local acidic environment**. We speculate that our synthesized Pt/MgO nanosheets can create a local acid-like environment under alkaline conditions. To test this speculation, we employed operando Raman spectroscopy and synchrotron radiation Fourier transform infrared spectroscopy (FTIR) under HER reaction conditions. These measurements will enable us to identify the key acidic intermediates of $H_3O^+$ of Pt/MgO during the HER process and to monitor their dynamic evolution. The operando measurements were conducted under three different conditions: open-circuit (immersed in KOH electrolyte), near the onset potential at a current density of $-0.5 \, mA \, cm^{-2}$ ($-10 \, mV$), and the overpotential at $10 \, mA \, cm^{-2}$ ($-40 \, mV$). Figure 2a shows the Raman spectra of Pt/MgO against the applied potential. Interestingly, when the applied potential is reduced to $-40 \, mV$ vs. RHE, a new peak at $\sim 1750 \, cm^{-1}$, which is assigned to $H_3O^+$ intermediate species, is observed[29], whereas the peak of $H_2O$ ($1600 \, cm^{-1}$) becomes weaker and the G-band of graphite ($1580 \, cm^{-1}$) remains unchanged, indicating the facilitated water dissociation on the surface of MgO and thereby the generation of abundant $H_3O^+$ intermediates. This observation is also supported by the cutting-edge operando synchrotron Fourier transform infrared (SR-FTIR) spectroscopy, which is highly sensitive to the functional groups within $\sim 3$ molecular monolayers. The FTIR spectra of Pt/MgO display a progressively intensified absorption band at $3525 \, cm^{-1}$ that is ascribed to the stretching vibrations of the O-H group in $H_3O^+$ when the applied potential is reduced to

$-40 \, mV$ vs. RHE (Fig. 3b and Supplementary Fig. 8)[30,31]; similar phenomena are observed on MgO nanosheets, but not on Pt/C (Supplementary Fig. 8), indicating the critical role of MgO in facilitating water dissociation and creating the $H_3O^+$ intermediates. More strikingly, the potential-dependent signal of $H_3O^+$ is reversible, manifesting that the local acid-like environment is formed only during the process of HER (Supplementary Fig. 9). In addition, it is worth noting that the detection limits of SR-FTIR spectroscopy and Raman spectroscopy are approximately ppm levels for hydroxyl groups or hydration molecules[32,33], which is much higher than that of the content of H+ ($\sim 10^{-14} \, M$) ionized from water in the 1 M KOH solution. Such a high level of $H_3O^+$ cannot be ubiquitous in KOH but can accumulate only within a local region. The generation of $H_3O^+$ species under specific alkaline conditions (pH = 13) was also discovered by Wang et al. on an ill-defined Pt site of a commercial Pt/C catalyst[29]; however, how to intentionally mediate and make good use of the $H_3O^+$ species remains a major challenge. Our aim in this work is to engineer this unique $H_3O^+$ species to create a desired local acidity under widespread alkaline conditions (Supplementary Fig. 10).

To obtain further information on the coordination and interactions between $H_3O^+$ and catalysts and to reveal the essential role of the Pt nanoparticles, we resorted to the operando XAFS measurements to observe the local coordination environment of the catalysts during the reaction. Operando XAFS is a very powerful tool for this purpose due to its element specificity and local structure sensitivity. Supplementary Fig. 11 shows the Pt L3-edge X-ray absorption near-edge structure (XANES) spectra of Pt/MgO under the operando conditions. A comparison of the XANES spectral features of Pt/MgO with those of Pt foil and PtO2 reveals that the Pt in Pt/MgO under the ambient conditions is in a metallic state, instead of a partially-oxidized state as reported for many other supported Pt nanoparticles[34–36]. In addition, the absence of Pt-O bonds in the $k^2$-weighted Fourier transformed (FT) EXAFS spectra further excludes the surface oxidation of the Pt particles in Pt/MgO (Supplementary Fig. 12). To magnify the spectral changes induced by the applied potentials and to identify the adsorbates on Pt, the XANES difference, or

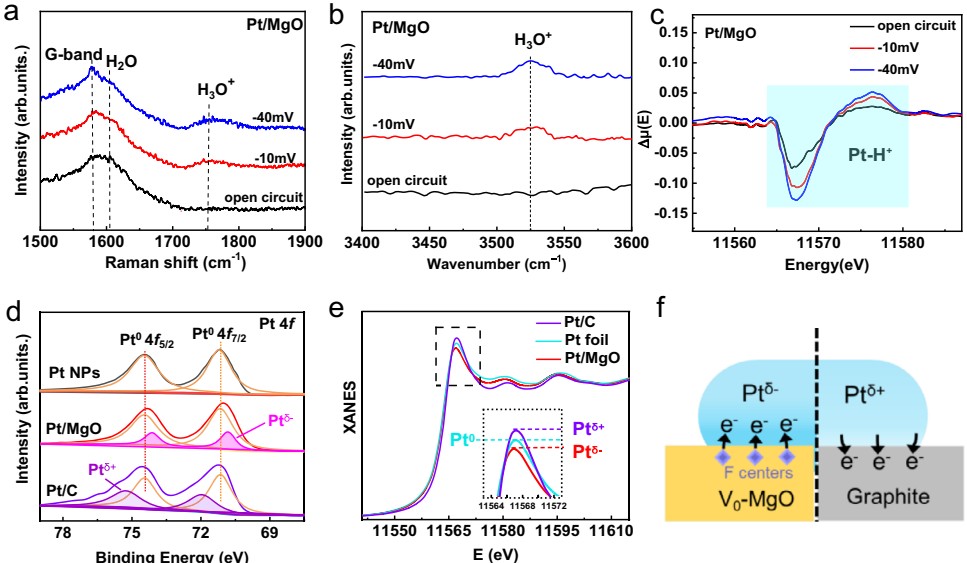

**Fig. 2 Operando spectroscopy characterization and chemical states of Pt.** The operando Raman spectra (**a**), SR-FTIR spectra (**b**), and XANES difference spectra (**c**) of Pt/MgO in 1 M KOH. **d** Pt 4$f$ XPS spectra and **e** Pt L$_3$-edge XANES spectra of Pt/MgO, Pt foil and Pt/C. **f** Graphical representations of the electron transfer between Pt nanoparticles and the support (V$_O$−MgO and graphite).

ΔXANES, analysis was conducted, and the ΔXANES spectra are shown in Fig. 2c. At various potentials, the overall ΔXANES profile resembles that for H-adsorbed metallic Pt but differs from that for OH-adsorbed Pt[37]. When the applied potential is reduced to −40 mV versus RHE, the magnitude of the ΔXANES results increases remarkably. This change is caused by the progressively increased adsorption coverage of H$_3$O$^+$ on Pt, indicating that the generated H$_3$O$^+$ intermediates are enriched in the proximity of Pt nanoparticles. Summarizing the above operando Raman, FTIR, and XANES spectra, we conclude that an abundance of H$_3$O$^+$ has been generated and forms a local H$_3$O$^+$ enriched, acid-like reaction environment around Pt nanoparticles in Pt/MgO.

To investigate the origin of the local acid-like environment created around these Pt nanoparticles, electron paramagnetic resonance (EPR), X-ray photoelectron spectroscopy (XPS), and XANES spectroscopy measurements were conducted on Pt/MgO in comparison to the Pt/C and MgO references. The EPR spectra of both Pt/MgO and MgO (Supplementary Fig. 13) show a strong signal at $g = 2.00$ arising from oxygen vacancies (V$_O$), suggesting a high concentration of V$_O$ in both the MgO and Pt/MgO. The abundance of oxygen vacancies as n-type doping could significantly improve the conductivity of MgO nanosheets. In addition, the oxygen vacancies of MgO are occupied by unpaired electrons (F centers), and the negative charge will transfer to the metal sites when the metal is trapped by the F centers, giving rise to the formation of electron-enriched Pt$^{\delta-}$ species[38]. This deduction is verified by the Pt 4$f$ XPS spectra as shown in Fig. 2d. The Pt 4$f_{7/2}$ XPS spectrum of Pt/MgO can be deconvoluted into two peaks at 71.4 and 70.6 eV. The 71. 4 eV peak is ascribed to the Pt$^0$ state as seen in Pt foil. The 70.6 eV peak in the lower binding energy region can arise only from negatively-charged Pt atoms (Pt$^{\delta-}$). More importantly, the Pt 4$f$ XPS spectra is highly sensitive to the surface, which excludes surface oxidation of the Pt particles in Pt/MgO. The formation of Pt$^{\delta-}$ in Pt/MgO is further confirmed by the Pt L$_3$-edge XANES spectra as shown in Fig. 2e. The main characteristic peak located at ~11568 eV in the Pt L$_3$-edge XANES spectra is the so-called white-line peak, which arises from an electron transition from the occupied Pt 2$p_{3/2}$ orbital to the empty 5$d$ orbital, and thus is indicative of the Pt 5$d$ occupancy. The slightly weaker white line of Pt/MgO relative to that of Pt foil thus

indicates the higher 5$d$ occupancy of Pt, in agreement with the existence of Pt$^{\delta-}$ therein. A similar phenomenon was also reported in TiO$_2$-supported metal catalysts due to strong metal-support interactions and in Pt-based alloys[39–42]. In contrast, in the commercial Pt/C, electron transfer occurs from Pt nanoparticles to the carbon support, as evidenced by XPS (Fig. 2d) and XANES (Fig. 2e) spectra showing the Pt$^{\delta+}$ states. All of these results suggest the strong electronic interactions between Pt nanoparticles and MgO at the interface, where electrons transfer from V$_O$ in MgO to the supported Pt nanoparticles (Supplementary Figs. 14, 15), resulting in the formation of negatively-charged Pt$^{\delta-}$ at the interface (Fig. 2f). Due to the electrostatic attraction between the negatively-charged Pt$^{\delta-}$ atoms and the positively-charged H$_3$O$^+$ species whose formation is facilitated by the V$_O$−enriched MgO, a local acid-like environment around Pt$^{\delta-}$ nanoparticles is created.

**Electrocatalytic performance toward HER.** From the above operando Raman, SR−FTIR, and XANES characterizations, it is evidenced that a local acid-like reaction environment is created around Pt$^{\delta-}$ nanoparticles in Pt/MgO, which is expected to be able to substantially boost HER activity. Thus, the electrocatalytic HER activity of Pt/MgO, commercial Pt/C, and MgO catalysts was evaluated in 1.0 M KOH. As a reference, the HER performance of commercial Pt/C was also evaluated under acidic conditions (the MgO is easily dissolved in acidic media). The Pt/MgO catalyst exhibits remarkable activity, with an overpotential of 39 mV at a current density of 10 mA cm$^{-2}$ (Fig. 3a), which is significantly lower than the 62 mV overpotential of 20 wt% Pt/C. Notably, the HER activity of the MgO support is almost negligible. To estimate the HER catalytic reaction kinetics, the Tafel slope is derived from the LSV curves, with values of 352, 68, and 39 mV dec$^{-1}$ for MgO, Pt/C, and Pt/MgO, respectively (Fig. 3b), indicating that Pt/MgO affords faster HER kinetics than Pt/C and follows the Volmer−Heyrovsky mechanism[43]. More importantly, by comparing the LSV curves and the Tafel slopes of Pt/MgO with those of Pt/C under acidic and alkaline conditions, one can find that the alkaline HER performance of Pt/MgO is close to the acidic HER behavior of Pt/C, suggesting the acid-like HER behavior of Pt/MgO in an alkaline medium. The EIS curves of Pt/MgO (Supplementary Fig. 16) show an electrochemical resistance

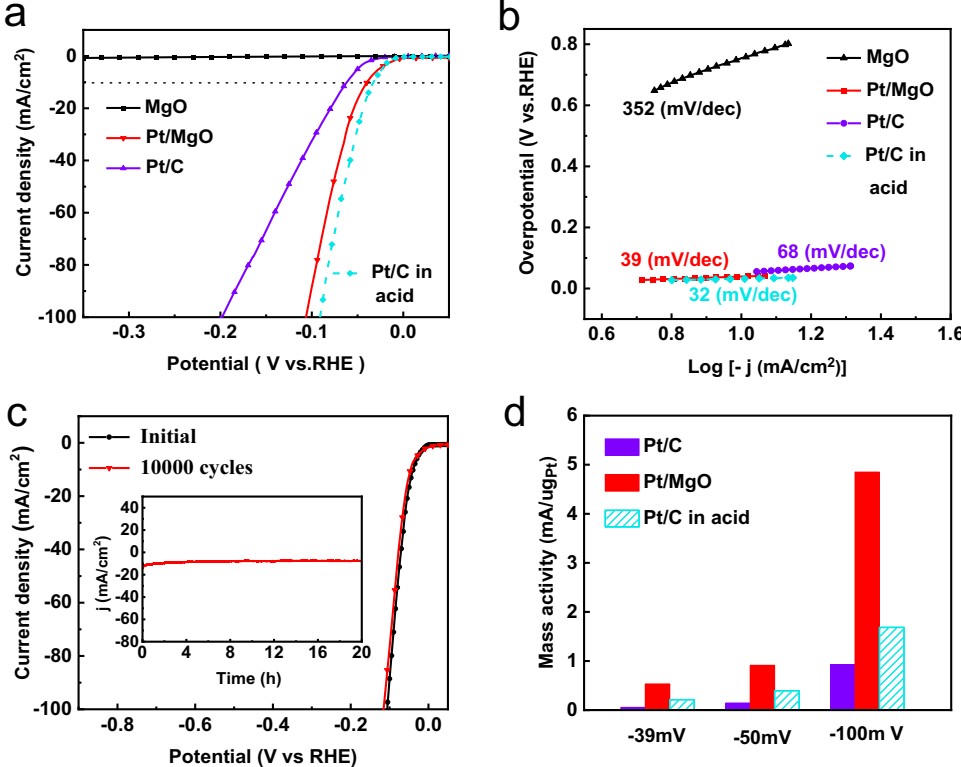

**Fig. 3 Electrocatalytic performance of Pt/MgO and reference samples for the HER. a** Linear sweep voltammetry (LSV) curves of Pt/MgO, Pt/C, and MgO in 1 M KOH, with that of Pt/C in $H_2SO_4$ obtained for comparison. **b** The corresponding Tafel plots were calculated from the LSV curves of Pt/MgO, Pt/C, and MgO. **c** The initial and 10,000th polarization curves recorded from Pt/MgO in 1 M KOH (inset: chronopotentiometric curve at a current density of 10 mA cm$^{-2}$). **d** Comparison of the mass activity of Pt/MgO and Pt/C under various conditions.

of 7 Ω, which is smaller than that of Pt/C (15 Ω), indicating the substantially facilitated interfacial electron-transfer kinetics in Pt/MgO. In addition, the durability and stability of Pt/MgO in 1.0 M KOH were implemented by an accelerated durability test (ADT) and chronoamperometry test as depicted in Fig. 3c. The ADT performance displays a slight drop after 10,000 cycles compared with the initial curve in 1.0 M KOH. Furthermore, the current density at 10 mA cm$^{-2}$ (the inset of Fig. 3c) of Pt/MgO remains virtually unchanged after 20 h of operation. More importantly, the TEM image (Supplementary Fig. 17) and XRD pattern (Supplementary Fig. 18) of Pt/MgO after continuous operation also maintain the original features, and the chemical states of Pt, Mg, and O as inferred from the XANES spectra (Supplementary Fig. 19) and XPS spectra (Supplementary Fig. 20) remain unchanged, confirming that Pt/MgO is stable in the long-term electrochemical process and that the negatively-charged Pt$^{\delta-}$ nanoparticles could not be oxidized during the HER. The EPR spectrum of Pt/MgO after the HER is also shown in Supplementary Fig. 21. It is obvious that the signal at $g = 2.00$ arising from oxygen vacancy still remains significant, suggesting the persistence of oxygen vacancies in Pt/MgO during HER. To further quantitatively compare the electrochemical activity of Pt/MgO with that of 20 wt% Pt/C, the mass activities at various potentials were determined, as illustrated in Fig. 3d. The mass current density of Pt/MgO reaches up to 0.53 mA µgPt$^{-1}$ at −39 mV vs. RHE, tenfold higher than that of 20 wt% Pt/C (0.05 mA µgPt$^{-1}$) in alkaline medium. Moreover, the ECSA-normalized HER polarization curves (Supplementary Fig. 22) and the turnover frequency (TOF) (Supplementary Fig. 23) suggest that the improvement in HER catalytic performance is mainly due to the increased catalytic activity of the intrinsic active sites. Reducing the catalyst loading on the cathodes of Pt/C and Pt/

MgO has no impact on the interaction between H* and the catalyst surfaces (Supplementary Fig. 23), further confirming the high intrinsic HER activity of Pt/MgO. It is worth noting that the mass current activity of Pt/MgO is 2.5-fold higher than that of similarly sized 20 wt% Pt/C (0.2 mA µgPt$^{-1}$) in an acidic medium. This performance is due to the existence of the electron-enriched Pt$^{\delta-}$ as will be discussed later. In addition, the Pt/MgO shows nearly 100% Faradaic efficiency in base (Supplementary Fig. 24). More importantly, a series of experiments and simulations, including the synthesis of negatively-charged Pt$^{\delta-}$ nanoparticles supported on TiO$_2$ (without strong water-dissociation ability), reducing the quantity of oxygen vacancies on MgO, and the calculations of the water-dissociation ability of negatively-charged Pt$^{\delta-}$ (Supplementary Figs. 25–29), show that the high HER activity comes from the synergistic effect of the local reaction environment and the negatively-charged Pt$^{\delta-}$, rather than only from the modified electronic structure of Pt. These results confirm that Pt/MgO possesses the high intrinsic HER activity, in accord with the expectations based on the local acidic environment. In addition, we synthesized other noble metal (Au, Ru, and Ir) nanoparticles supported on MgO to replace Pt (Supplementary Figs. 30–33). Compared with that of Ru/C and Ir/C, the HER performance of Ru/MgO and Ir/MgO is much better, indicating that an abundance of H$^+$ created by MgO can also improve the alkaline HER activity of other noble metals. Structural characterizations show that negatively-charged metal sites are also formed on the Ir surface (Supplementary Fig. 34). Nevertheless, the HER performance of Ir/MgO is still not as good as that of Pt/MgO (Supplementary Fig. 35), probably due to the inherent differences between Pt and Ir such that Pt has the most approachable zero hydrogen absorption energy.

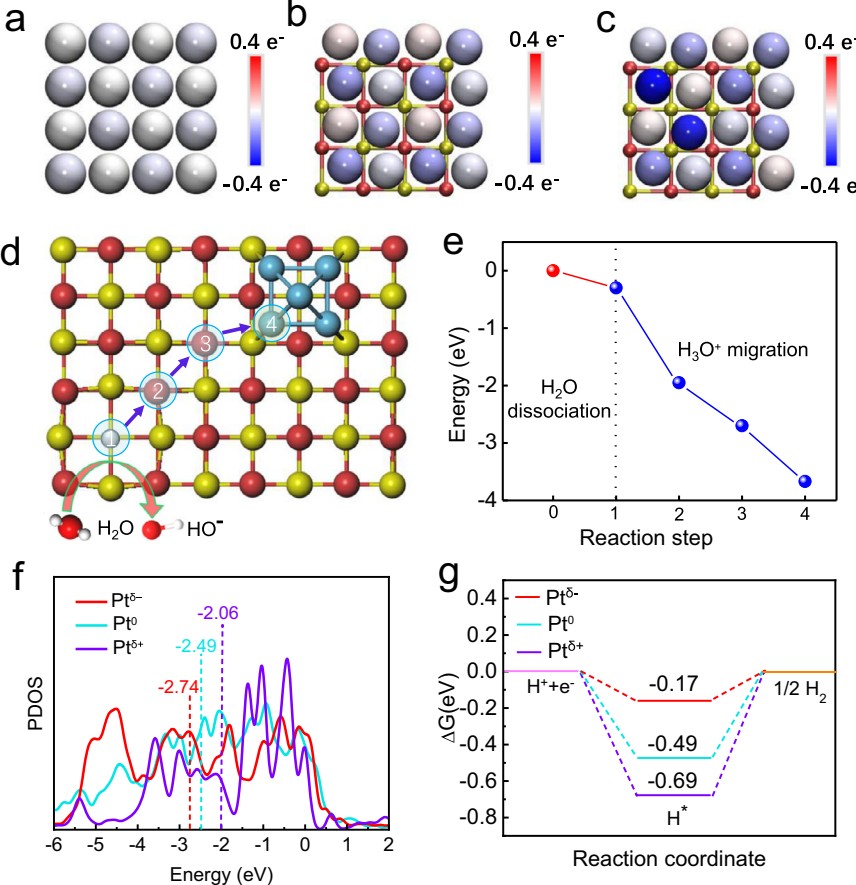

**Fig. 4 Theoretical investigations.** Atomic charge distribution of Pt (**a**), Pt-MgO (**b**), and Pt-MgO-Vo (**c**). The balls in red, yellow, and gray represent O, Mg, and Pt atoms, respectively. Schematic illustration (**d**) and energies resulting (**e**) from processes of adsorption of $H_2O$ followed by its dissociation and proton migration on the MgO (001) surface with anionic vacancies. **f** Calculated PDOS of the Pt d orbital of $Pt^{\delta-}$, $Pt^0$, and $Pt^{\delta+}$. **g** Calculated adsorption energies of H on the surface of $Pt^{\delta-}$, $Pt^0$, and $Pt^{\delta+}$.

**Density functional theory (DFT) calculations.** Based on the aforementioned operando spectroscopic observations that MgO can create a local acid-like reaction environment around $Pt^{\delta-}$, DFT calculations were conducted to understand how the oxygen vacancies in MgO influences the charge density distribution of supported Pt and how the various electronic structures of Pt species affects the local reaction environment and HER activity. As shown in Fig. 4a–c, compared with the cases of Pt and Pt supported on oxygen vacancy-free MgO, in Pt supported on oxygen vacancy-rich MgO, strong electron injection occurs from oxygen vacancies occupied by unpaired electrons (F centers) into Pt nanoparticles, which leads to the formation of electron-enriched Pt species ($Pt^{\delta-}$). The Bader charge analysis results further shows that approximately $0.3e$ electrons are transferred from oxygen vacancy-rich MgO to the Pt atoms adjacent to the oxygen vacancy, and no significant electron transfer occurs between oxygen vacancy-free MgO and Pt nanoparticles. Next, we explored the dissociation of $H_2O$ molecules followed by a proton migration process on the MgO (001) surface as shown in Figs. 4d, 5e. A water molecule is dissociated into an OH* and a chemisorbed H* species, with an energy barrier of −1.7 eV on the MgO (001) surface, which is significantly lower than that on Pt (0.06 eV)[44], $RhO_2$ (0.35 eV)[45], and so on (Supplementary Fig. 36). This finding is demonstrated by the carbon monoxide (CO) stripping tests (Supplementary Fig. 37). The CV curves show that the stripping peak for $CO_{ad}$ oxidation of Pt/MgO has a more negative value than that for Pt/C, indicating a better water-dissociation ability for Pt/MgO over that of Pt/C[29]. The negligible

energy barrier for breaking the OH-H bonds suggests that the water-dissociation step on MgO is energetically favorable on Mg sites. The large distance between H* adsorbed on the lattice $O^{2-}$ atoms and OH* on the lattice $Mg^{2+}$ atoms stabilizes the dissociated structure due to the weak intermolecular interactions. Next, the free energy diagram was constructed to gain further thermodynamic insight into the migration process of H+ dissociated from water molecules, as shown in Fig. 4d, e and Supplementary Fig. 38. Along the path indicated by the arrows, the migration energy of H+ decreases gradually, indicating the strong tendency of the H+ to accumulate around $Pt^{\delta-}$ nanoparticles. Considering the fact that $V_O$-rich MgO creates a local acid-like environment for negatively-charged $Pt^{\delta-}$, we calculated the Gibbs free energy of hydrogen adsorption ($\Delta G_{H*}$), which is widely accepted as a universal descriptor for the HER in acid, to judge the activity of Pt in the step of hydrogen adsorption. We examined the effects of three different charge states ($Pt^{\delta-}$, $Pt^0$, and $Pt^{\delta+}$) of the Pt nanoparticles on the HER activity (Supplementary Fig. 39). The calculated partial density of states (PDOS) of Pt d orbitals with different charge states are shown in Fig. 4f, where the d-band center position is explicitly labeled. Apparently, $Pt^{\delta-}$ has the lowest d-band center position (−2.74 eV), as compared with that of $Pt^0$ (−2.49 eV) and $Pt^{\delta+}$ (−2.06 eV). According to the d-band theory, a lower d-band center (farther away from Fermi level) of metal-based catalysts leads to weaker interaction between the 1s orbital of hydrogen and the unfulfilled d orbitals of the metals[46]. To verify this point, we calculated the Gibbs free energy ($\Delta G_{H*}$) of H* adsorption on the surfaces of $Pt^{\delta-}$, $Pt^0$, and

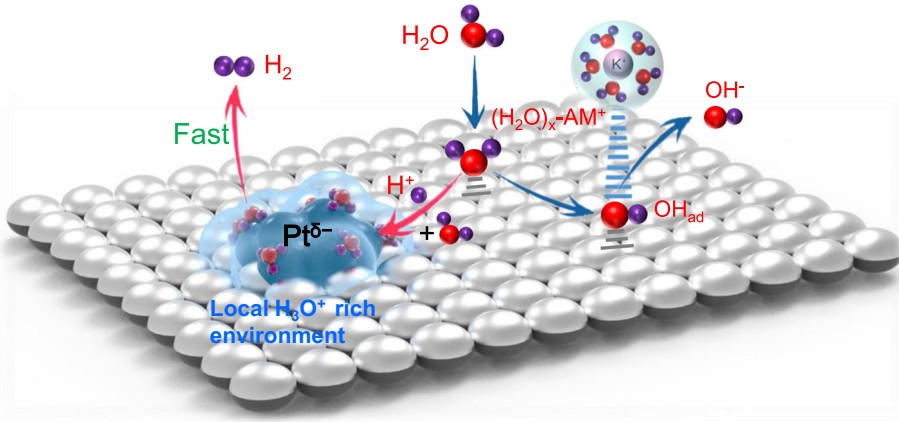

**Fig. 5 Reaction mechanism.** Schematic representation of water dissociation, formation of $H_3O^+$ intermediates, and subsequent formation of $H_2$ as well as $OH^-$ desorption on the Pt/MgO nanosheets.

$Pt^{\delta+}$ (Fig. 4g). $\Delta G_{H*}$ is known as a reasonable descriptor of the HER activity, and the optimum value of $|\Delta G_{H*}|$ should be close to zero, as inferred from the Sabatier principle[47]. The calculated $\Delta G_{H*}$ of $Pt^{\delta-}$ ($-0.17$ eV) is closer to 0 eV than that of $Pt^0$ ($-0.49$ eV), and $Pt^{\delta+}$ ($-0.69$ eV), confirming a more favorable H* desorption (Supplementary Fig. 40), thus boosting the HER activity.

Based on the above experimental and theoretical results, we propose an HER mechanism for the $Pt^{\delta-}$ nanoparticles in an alkaline electrolyte (Fig. 5). The oxygen vacancy-rich MgO facilitates the water dissociation and the surface $Pt^{\delta-}$ species promote $H_3O^+$ migration and accumulation to achieve the acid-like HER activity and kinetics in alkaline media. The specific reaction mechanism is as follows: First, a water molecule diffuses to an oxygen vacancy site of the MgO surface, where it is adsorbed and dissociated into an OH* and an H*. As the water-dissociation reaction proceeds, a great many H* species are generated which subsequently combine with adjacent $H_2O$ molecules to form $H_3O^+$. Due to the lower migration energy for the $H^+$ and the additional electrostatic attraction between the positively-charged $H_3O^+$ and the negatively-charged $Pt^{\delta-}$, the $H^+$ easily migrates throughout the rigid water network of the double layer caused by the positive shift in the potential of zero free charge (pzfc) to form a local acid-like environment around $Pt^{\delta-}$. Within the framework of pzfc, the local acid-like environment will shift the pzfc of Pt toward the HER equilibrium potential, and then promote the hydrogen evolution reaction by lowering the energy barrier. Benefiting from the local acid-like reaction environment, afterward, $H_3O^+$ then adsorbs hydrogen atoms H* on the $Pt^{\delta-}$ surface and later combines with another H* to generate $H_2$ following the Tafel step. Meanwhile, the remaining OH* species in the oxygen vacancy will combine withs a hydrated $K^+$ ion to form a hydroxyl-water-cation ($OH_{ad}$-$(H_2O)_x$-$K^+$) adduct. The existence of $OH_{ad}$-$(H_2O)_x$-$K^+$ was confirmed by the deuterium ($^2H$, D) nuclear magnetic resonance (NMR) and CO stripping experiments (Supplementary Fig. 41). Then, the adduct quickly transfers to the electrolyte across the electrical double-layer, causing no change in the local $OH^-$ concentration[24]. After the release of OH*, the oxygen vacancy is restored, and another water molecule diffuses to the oxygen vacancy site and continues the next reaction cycle. As a result, the $V_O$-rich MgO/Pt catalyst can continuously and stably produce and accumulate a large amount of $H_3O^+$, thus creating a local acid-like environment. The synergistic effect of the local acid-like reaction environment and electron-enriched $Pt^{\delta-}$ species endow the catalyst with excellent intrinsic HER activity.

## Discussion

In summary, we have developed a facile pathway to create a local acid-like reaction environment in an alkaline medium. Taking the $V_O$-MgO nanosheet supported Pt as a prototypical example, this catalyst is favorable for water dissociation and the formation of electron-enriched $Pt^{\delta-}$ species. A combined study of operando Raman, SR-FTIR, and XAFS spectra evidences that negatively-charged $Pt^{\delta-}$ species accelerate positively-charged $H_3O^+$ migration and accumulation, resulting in a local acid-like environment. This acid-like environment provides Pt with a favorable reaction condition for the HER in the alkaline electrolyte. As a result, the Pt/MgO catalyst shows an overpotential of 39 mV at a current density of 10 mA cm$^{-2}$, which is significantly lower than the 62 mV of 20 wt% Pt/C in an alkaline medium and close to the acidic HER behavior of Pt/C (33 mV). Our study provides insight into tailoring the local reaction environment to design high-performance electrocatalysts in a more rational and precise way.

## Methods

**Material characterizations.** The TEM images, the HRTEM, and EDS mapping analyses were performed on a JEM-2100F field emission transmission electron microscope at an acceleration voltage of 200 kV. XPS spectra were acquired on an ESCALAB MKII with Mg Kα (hυ = 1253.6 eV) as the excitation source and were corrected for specimen charging by referencing C 1s to 284.5 eV. The EPR measurements were performed in a JSE-FA200 EPR spectrometer at X-band (~9 GHz) with a resolution of 2.35 μT at 300 K. The Pt L-edge X-ray absorption near-edge (XANES) spectra were measured at the 1W1B beamline of Beijing Synchrotron Radiation Facility (BSRF). The O K-edge XANES spectra were collected at the BL12B beamlines of NSRL (Hefei, China), respectively.

**Synthesis of MgMOF-74 precursor.** Mg(NO$_3$)$_2$·6H$_2$O (0.4 g) and 2,5-dihydroxy benzene carboxylic acid (0.09 g) were mixed and dissolved in 40 mL of N,N-dimethylformamide (DMF). A mixture of ethanol, water, and triethylamine mixtures (9:9:1, v/v/v) was added to the above solution and stirred for 3 h at room temperature. The resulting MgMOFs were collected by centrifugation and then redispersed in DMF. The suspension was treated at 100 °C for 2 h to dissolve other amorphous impurities.

**Synthesis of Pt/MgO and MgO.** In a typical procedure, a mixture of 50 mg of MgMOF and 3 mL of aqueous H$_2$PtCl$_6$ solution (10 mg/mL) was dissolved in 8 mL of deionized water and ultrasonically treated for 10 min. After magnetic stirring for 24 h at room temperature, the precipitates were collected by centrifugation and then freeze-dried to obtain the precursor. The precursor was then placed into a tube furnace and heated to 700 °C in a mixture of oxygen and nitrogen (O$_2$:N$_2$ = 5:95) for 4 h at a heating rate of 10 °C/min to obtain the desired Pt/MgO. MgO was synthesized in a similar way without adding an aqueous H$_2$PtCl$_6$ solution.

**DFT calculation details.** The DFT calculations were carried out with the Quantum Espresso software package[48]. The Generalized gradient approximation (GGA) in the Perdew–Burke–Ernzerhof (PBE) parametrization and projected augmented

wave (PAW) method were used to describe the electron exchange-correlation and electron-ion interaction. DFT-D3 scheme was adopted to consider the long-range van der Waals interaction. The kinetic energy cutoffs of the plane wave and electron density were set as 80 and 500 Ry. The energy and atomic force convergence criteria of $1.0 \times 10^{-3}$ meV/atom and 0.05 eV/Å was ensured for the self-consistent field calculations and structural optimizations. The atomic changes were calculated with Bader's analysis using the fast algorithm developed by G. Henkelman[49].

The dissociation of water molecules and migration of $H_3O^+$ were simulated with a four-layer MgO[001] slab model (containing 96 Mg atoms and 96 O atoms) with a $Pt_5$ cluster attached to the surface. The two bottom layers were fixed to simulate the bulk MgO. A surface O atom was removed to simulate the O vacancy.

The $Pt^{\delta-}$, $Pt^0$, and $Pt^{\delta+}$ model catalysts were simulated with a Pt[100] @MgO[001] (with one O vacancy) slab model, a Pt[100] slab model, and a $Pt_4$ cluster model on a $6 \times 6$ graphene superlattice.

We have used two models for different purposes. (1) For the simulations of the dissociation of water on the surface of MgO and the following migration of proton towards Pt nanoparticles, the model should contain a MgO surface, O vacancies, and a Pt cluster. Hence a "MgO surface-O vacancy-$Pt_5$" model was built for these simulations. We have to use a much smaller $Pt_5$ cluster to represent the Pt nanoparticle of 5 nm because a larger Pt cluster would cause a much larger size model which is beyond our computational resources. Actually, the model with the $Pt_5$ cluster on MgO already contains 199 atoms. Besides, this simulation focuses on the production and migration of protons. Since the small Pt cluster's affinity for proton is close to that in the large Pt nanoparticles, this approximation should be acceptable. (2) Another model is the "MgO[001]-Pt[100] interface" model, which is used to study the interfacial charge transfer between MgO and Pt nanoparticles. As you commented, the quantum confinement could dramatically affect the position and occupation of energy levels of small clusters, including the Fermi level, which is crucial for charge transfer. Hence the layered slab models are widely adopted for interfacial charge transfer simulations instead of the cluster model.

**Electrochemical measurements**. All electrochemical measurements were performed on a CHI760E electrochemical workstation using a three-electrode system in a 1 M KOH electrolyte. Generally, 5 mg of catalysts were dispersed in a 1 ml mixture solution of water and ethanol (Vwater/Vethanol = 3/1) followed by the addition of 30 μL Nafion solution. Subsequently, the mixed suspension was sonicated to form a homogeneous ink. Five microliters of the catalyst ink was dripped onto a glassy carbon electrode (3 mm in diameter). The coated glassy carbon electrode was used as the working electrode and we use a saturated Ag/AgCl electrode and a graphite rod as the reference electrode and counter electrode, respectively. Before the electrochemical measurements, the cyclic voltammetry was recorded with a scan rate of 50 mV s$^{-1}$ for several cycles until it is stable. In all measurements, linear sweep voltammetry (LSV) tests were carried out with iR compensation and the presented potential values were calibrated to a reversible hydrogen electrode (RHE). E(RHE) = E(Ag/AgCl) + 0.198 V + (0.059 × pH)V. The stability of catalysts was conducted using a chronoamperometry method at a current density of 10 mA cm$^{-2}$. Electrochemical impedance spectroscopy (EIS) measurements were measured in the frequency range of 0.1 to $10^5$ Hz.

**Operando SR-FTIR measurements**. Operando synchrotron radiation FTIR data were collected at the infrared beamline BL01B of the National Synchrotron Radiation Laboratory (NSRL, China) through a homemade cell. The catalysts were coated on the carbon paper ($1 \times 2$ cm$^2$) as the working electrode, which was tightly pressed against the ZnSe crystal window with a micro-scale gap to reduce the loss of infrared light. The FTIR tests were measured by reflection mode to guarantee the good signal of SR-FTIR spectra. Before data collection, a voltage was applied to the catalyst electrode for 10 min. During operando FTIR measurements, the background spectrum of the working electrode was obtained at an open-circuit voltage before HER measurement. In the end, we dealt with the infrared data using OPUS software to get better signal-to-noise spectra.

**Operando Raman measurements**. Operando Raman measurements were performed on the LabRamHR Raman Spectrometer (laser wavelength = 532 nm). The catalyst ink was dropped on a rough Au electrode as the working electrode. Before tests, the working electrode was first infiltrated into the electrolyte. Similarly, we performed in situ electrochemical tests through a homemade cell to obtain better signal data. Before data collection, a voltage was applied to the catalyst electrode for 20 min.

**Operando XAFS measurements**. The Pt L$_3$-edge (11,564 eV) XAFS spectra were measured at the 1W1B beamline of Beijing Synchrotron Radiation Facility (BSRF), China. Operando XAFS measurements were performed in alkaline solution by using a smart homemade cell. The XAFS spectra were collected through the solid-state detector to obtain weak signals in the electrochemical reaction process. The catalysts was uniformly and stably distributed over carbon paper as the working electrode, and the back of the carbon paper is fixed with Kapton film to ensure that all electrocatalysts can react with the electrolyte. In order to monitor the changes in the HER process, the cathode voltage of 0 to $-0.04$ V was applied for 10 min as the pretreatment step.

## Data availability

All data generated in this study are provided in the Supplementary Information/Source Data file. Source data are provided with this paper.

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

## Acknowledgements

This work was financially supported by the National Key Research and Development Program of China (2021YFA1600800, W.Y., 2021YFA1500403, Z.S.), the National Natural Science Foundation of China (Grants No. 11975234, W.Y., 11775225, W.Y., 12075243, Z.S., 12005227, H.T., and U1932211, W.Y.), the Fundamental Research Funds for the Central Universities (WK2310000103, C.W.), the Users with Excellence Program of Hefei Science Center CAS (2021HSC-UE002, C.W. and 2021HSC-KPRD002, W.Y.), the Innovative Program of Development Foundation of Hefei Center for Physical Science and Technology (2020HSC-CIP013, W.Y.), this work was partially carried out at the USTC Center for Micro and Nanoscale Research and Fabrication. The authors would like to thank BSRF, SSRF, and NSRL for the synchrotron radiation beamtime.

## Author contributions

H.T., Z.S., and W.Y. designed the study. H.T., B.T., Y.L., Q.J., and L.L. carried out the sample synthesis and electrochemical measurements. B.T., Y.L., N.L., and Y.W. performed the TEM, XRD, and EPR characterization. C.W. finished the DFT calculations and analysis. H.T., Y.L., Q.J., F.H., and L.L. carried out the in situ and ex situ XAFS, in situ Raman, and infrared experiments. H.D., S.F., and Z.L. carried out the XANES and SEM characterizations. H.T., Z.S., and W.Y. wrote the manuscript. The other authors provided reagents and performed some of the experiments.

## Competing interests

The authors declare no competing interests.
