## [Peer Review File · Nature Communications]

Engineering a local acid-like environment in alkaline medium for efficient hydrogen evolution reactionREVIEWER COMMENTS

Reviewer #1 (Remarks to the Author):

The authors combined Pt nanocatalysis and oxygen-vacancy rich MgO to design catalysts for hydrogen evolution under high pH conditions. The author used the results presented in Ref. 24 of the acid-like environment in the double layer region of Pt nanocatalysis under high pH conditions. The authors demonstrate that such an environment is also possible for oxygen-vacancy-rich MgO surfaces under high pH conditions. By combining Pt nanocatalysis and oxygen-vacancy rich MgO, the authors argue that a better catalyst for hydrogen evolution under high pH conditions was prepared. The work is original in the sense of combining the materials. However, the author shows that the use of oxygen-vacancy-rich MgO modifies the electronic properties of Pt nanocatalysis. Therefore, it is not clear if the better catalyst is the result of using the oxygen-vacancy rich MgO to create the acid-like environment or the results of the MgO modifying the electronic properties of the Pt nanocatalysis. The acid-like environment is also present in Pt/C under high pH conditions, without needing MgO. If the acid-like environment coming from MgO plays a pivotal role for the better catalyst, would it be possible to use other metals in place of Pt? Would it be possible to reduce the amount of oxygen-vacancy in MgO and study its effects on the catalysis of Pt/MgO?

Without clear evidence that the acid-like environment coming from MgO is the key for the better catalyst and not just the modification of Pt by substrate effects, the work is not showing a new design route.

The DFT calculations included by the author were performed using standard methodologies. However, details are missing about how the results presented in Fig. 5a to 5c were acquired; how was the atomic charge difference calculated, is the difference between what? The authors mention Pt nanoparticles in the test, but Fig 5a to 5c looks like an overlayer of Pt over MgO, and no Pt nanoparticles on MgO; is the Pt overlayer on MgO strained? The authors included calculations with a cluster of Pt₅ in Fig 5d. Charge analysis for such cluster on MgO systems will be more relevant here, even though the experimentally prepared nanoparticles have sizes of 5 nm; a cluster of Pt₅ has less than 0.5 nm diameter and quantum confinement effects are present.

Reviewer #2 (Remarks to the Author):

The authors have synthesized a Pt/MgO catalyst using a MOF-derived MgO support for alkaline HER. The authors claim that the as-prepared Pt/MgO generates a local acid-like environment in the alkaline

medium providing enhanced activity. They further show that vacancy-rich MgO nanosheets initiate the water dissociation and produce H_3O^+ , which finally accumulate around Pt δ^- to evolve H_2 . Although the concept is not novel, it is important to note that the authors have used several advanced operando techniques to carry out this work. Nonetheless, there are some major inconsistencies (see below) in the manuscript that should be addressed before it can be considered for publication in a highly reputed journal like Nature Communications.

1. First of all, why do the authors see a graphite peak in Figure 3a (Raman spectra) if they recorded the spectra only over Pt/MgO? Moreover, the experimental section also does not provide any details on the carbon contamination or the percentage of carbon left in Pt/MgO. I assume that the G band might be originating from (unremoved) carbon. If it is true, then this peak should show a broader nature at reduced potentials as the amount of adsorbed water will be inflated with potential. However, the figure shows quite the opposite (although the authors claim that it is unchanged).

2. It is quite a strange observation that MgO is responding for H_3O^+ (Supplementary Fig 7) although the proton adsorption ability of MgO is not adequate as compared to Pt. This is the reason the current density value for MgO is significantly low. Please incorporate additional evidence to prove this point.

3. If it is assumed that in Pt/C composite Pt is partially positively charged, then why do the authors observe H_3O^+ signal in SR-FTIR (Supplementary Fig 9)? Also, the pH effect on the local acidic medium is not logical. With pH, the adsorption of proton increases linearly. The signal intensity for H_3O^+ is higher (if it is not a spectral artifact) in pH 12 as compared to pH 13. In contrast, the Raman spectra did not give any signal for H_3O^+ (Supplementary Fig 7). Besides, this claim does not correlate with the study done in Nat. Commun. 10, 4876, 2019.

4. In theoretical calculation, the calculated energy for adsorbed proton (H^*) on Pt0 is not close to its well-known reported value. (see Chem. Sci., 2019, 10, 9165-9181). Also, do the authors have any evidence for hydroxyl-water-cation adduct.?

5. The XPS shift for Mg after inclusion of Pt looks empirical (supplementary Fig. 13). The figure reads more than a 0.7 eV shift which is probably uncorrected from the standard carbon profile. In addition, the authors should note that the comparison of Pt 4f spectra between Pt foil and Pt/MgO may not provide conclusive data (Fig. 3d) because the size of the Pt in the prepared catalyst is sufficiently small even though it belongs to the bulk Pt. Ideally, Pt nanoparticles with similar sizes should be used for comparison.

6. What do the authors mean by “after reaction” in supplementary Fig 8? How is it correlated with reversibility?

7. What was the main reason to use MOF for the synthesis? Is it because of high surface area or to generate vacancies?

8. MgO could form Mg(OH)₂ in KOH solution (at least at the surface). An X-ray powder diffraction of the catalysts before and after the reaction would help to get more insights.

9. What is the pH value used for overpotential calculation? and how this pH value was measured?

10. How does the electrochemical surface area influence the catalytic activity of Pt/C and Pt/MgO? Similarly, the Faradaic efficiency of the reaction should be measured

11. Authors should simulate the EXAFS spectra

Reviewer #3 (Remarks to the Author):

The authors demonstrate a methodology to tune the local reaction environment such that a local acid like environment is created in an alkaline medium which results in a catalyst surface with a superior HER performance. The catalyst synthesized is promising with an overpotential of 39 mV at 10 mA/cm² close to acidic HER activity of Pt/C which has an overpotential of 33 mV at a similar current density. Although the data presented is promising, the manuscript lacks in certain areas particularly in providing an explanation on some of the controversy surrounding alkaline HER reaction. I do think this work is significant to the field but some clarifications are necessary. There should be minor revisions based on the following questions and comments before consideration for publication.

1) I do not understand the purpose of Supplementary Fig 9, it shows H₃O⁺. What is the difference between Supp Fig 9 and Supp Fig 7d? What is different in how the measurement is taken?

2) Incorrect labelling of Fig. 6 (labelled as Figure 5)

3) Can the authors comment on the OH binding strength of their catalyst? The local reaction environment may not be the only thing that is altered and there might be other factors in play as well. There was a recent study by Marc Koper that talks about the role of adsorbed hydroxide in HER. There are other studies as well that discuss the role of adsorbed hydroxide. Because of the extensive discussion on adsorbed hydroxide in literature, I feel that it is important to discuss how hydroxide binding strength is changing with MgO compared to Pt/C and if it has any role.

Link to articles:

<https://www.nature.com/articles/s41560-020-00710-8> (The role of adsorbed hydroxide in hydrogen evolution reaction kinetics on modified platinum)

<https://pubs.acs.org/doi/abs/10.1021/acscatal.7b02787> (Adsorbed Hydroxide Does Not Participate in the Volmer Step of Alkaline Hydrogen Electrocatalysis)

4) The authors argue that the negatively charged Pt creates a local acid-like reaction environment. There are many studies in literature that talk about the potential of zero free charge (pzfc) of Pt in acid vs base. Most of the studies point out that the surface is more negative in base because the pzfc shifts towards more positive potentials. Recently a study by a group at MIT tracked electric fields (link (a) below) and showed this as well. There is a discussion about a negatively charged surface in base creating a more rigid water network as it interacts strongly with the electric field and is therefore more difficult to reorganize during the charge transfer through the double layer as shown by Marc Koper (link (b) below). I would really like the authors to comment on this and maybe include this in the manuscript as it is highly relevant. It could be that MgO acts just like Ni(OH)₂ clusters to improve activity rather than having any other effect.

a) <https://pubs.acs.org/doi/abs/10.1021/jacs.9b05148> (Tracking Electrical Fields at the Pt/H₂O Interface during Hydrogen Catalysis)

b) <https://www.nature.com/articles/nenergy201731> (Interfacial water reorganization as a pH-dependent descriptor of the hydrogen evolution rate on platinum electrodes)

5) The XPS and XANES for Pt/C in fig 3d and 3e might show that the surface is positively charged but under applied potential at HER relevant potentials this is highly unlikely, and I would think that the surface would be negatively charged at those potentials considering the information we have on the pzfc of the Pt surface. Can the authors comment on how they think this result is relevant at HER relevant potentials?

6) Gibbs free energy of hydrogen adsorption (ΔG_H^*) is a reasonable descriptor in acid but in base it gets more complicated which is why there is a lot of discussion on trying to identify an accurate descriptor in base. While it is still important to optimize hydrogen adsorption, most consider it an inaccurate descriptor in base just because it cannot explain everything. This study lacks any literature study about the different mechanisms in base which may be important to provide context to this study.

7) In supplementary Fig. 23, can the authors comment on the reproducibility of the CV in 1 M KOH. Generally, in my experience I have seen (110) and (100) peaks appear quite clearly in the CVs in acid and base. Are the CVs shown after a steady state is reached and there is no more change? Could the authors comment on the reproducibility of the results including the CVs? It would also be useful to know what hardware is used for KOH results. Whether it is glass or FEP based? How is it cleaned before tests?

Response to Reviewers and a Summary of Changes Made

Many thanks to the reviewers for having given us valuable comments on the manuscript of NCOMMS-21-31196A submitted to the journal of *Nature Communications*,

Title: Engineering a local acid-like environment in alkaline medium for efficient hydrogen evolution reaction

Authors: Hao Tan, Bing Tang, Ying Lu, Qianqian Ji, Liyang Lv, Hengli Duan, Na Li, Yao Wang, Sihua Feng, Zhi Li, Chao Wang, Fengchun Hu, Zhihu Sun and Wensheng Yan

We are grateful to the reviewers for their comments which help us to further improve the quality of our manuscript. At first, we are grateful to the reviewers for their positive comment on this work that “*The work is original in the sense of combing the materials...*” (Reviewer 1), and “*it is important to note that the authors have used several advanced operando techniques to carry out this work.*” (Reviewer 2), and “*I do think this work is significant to the field but some clarifications are necessary. There should be minor revisions based on the following questions and comments before consideration for publication.*” (Reviewer 3). The reviewers also presented some comments and suggestions for the improvement of this manuscript before it could be reconsidered for publication. We thank all reviewers for their careful and positive reviews of our manuscript. We have seriously addressed their questions and comments and have further improved our manuscript. The detailed reply to the comments will be presented in a point-to-point manner as follows.

The entire comments from 3 Reviewers

Reviewer #1: (Remarks to the Author)

The authors combined Pt nanocatalysis and oxygen-vacancy rich MgO to design catalysts for hydrogen evolution under high pH conditions. The author used the results presented in Ref. 24 of the acid-like environment in the double layer region of Pt nanocatalysis under high pH conditions. The authors demonstrate that such an environment is also possible for oxygen-vacancy-rich MgO surfaces under high pH conditions. By combining Pt nanocatalysis and oxygen-vacancy rich MgO, the authors argue that a better catalyst for hydrogen evolution under high pH conditions was prepared. The work is original in the sense of combing the materials. However, the author shows that the use of oxygen-vacancy-rich MgO modifies the electronic properties of Pt nanocatalysis. Therefore, it is not clear if the better catalyst is the result

of using the oxygen-vacancy rich MgO to create the acid-like environment or the results of the MgO modifying the electronic properties of the Pt nanocatalysis.

Reply: We sincerely appreciate the reviewer's insightful and professional comments. Those comments are all very valuable and helpful for us to revise and improve this manuscript. To illustrate the reviewer's concern regarding the real active species for the better HER performance, additional comparative experiments, electrochemical tests and theoretical calculations were carried out according to the reviewer's suggestions.

First of all, we would like to address that the most important achievement in this work is the deliberate creation of a desired local acid-like environment for the active sites of negative-charged $\text{Pt}^{\delta-}$ during the alkaline HER reaction. It is the synergy of the local acid-like reaction environment and the optimized active $\text{Pt}^{\delta-}$ sites that contributes to the better HER performance. First, multiple physicochemical interactions between the substrate, metal active sites and reaction intermediates create a beneficial local acid-like environment for the active sites, which has been clearly demonstrated by the in-situ Raman, synchrotron radiation Fourier transformed infrared spectroscopy (SR-FTIR) and X-ray absorption near-edge structure (XANES) spectroscopy. Second, the optimized active $\text{Pt}^{\delta-}$ sites are favorable for the subsequent H^* reactants and thus boosts the HER activity.

In order to further prove that the high activity originates from the synergistic effect of local acid-like reaction environment and negatively-charged $\text{Pt}^{\delta-}$, we have conducted a series of control experiments and theoretical calculations to exclude the possibility of high HER activity only originating from the MgO that modifies the electronic properties of the Pt nanoparticles. These include the synthesis of negatively Pt nanoparticles supported on TiO_2 (without strong water dissociation ability), reducing the amount of oxygen vacancy in MgO by using the acid etching method, and the calculations of the water dissociation ability of negative valence $\text{Pt}^{\delta-}$. The detailed answers are given in Reply to Question #3 below. Besides, all the insightful suggestions the reviewer raised have been thoroughly considered, and the corresponding revisions have been made. Again, we truly appreciate the reviewers' careful review and kind guidance on this work, which is vital for us to improve quality of this manuscript.

1)The acid-like environment is also present in Pt/C under high pH conditions, without needing MgO. If the acid-like environment coming from MgO plays a pivotal role for the better catalyst, would it be possible to use other metals in place of Pt?

Reply: We gratefully thank the precious time that the reviewer spent on making the constructive remarks, and we totally understand the reviewer's concern. First of all, we

would like to apologize for the mistake we made in Supplementary Fig 9 in the original manuscript, it is actually for the sample of Pt/MgO rather than for the sample of Pt/C. We are sorry for this typo. This can be inferred from the sentence in the main manuscript: “It is our aim **in this work** to engineer this unique H_3O^+ species for creating a desired local acidity under widespread alkaline conditions (**Supplementary Fig.9**).”.

As for the possibility of using other metals instead of Pt, we have tested other noble metals, such as Au, Ru and Ir supported on MgO, to replace Pt. By using the same synthesis steps, the Au/MgO, Ru/MgO and Ir/MgO catalysts were obtained and their XRD patterns are shown in Figure R1 below. We also took their TEM images. Unfortunately, the size of the Au particles is too large (~100 nm), and therefore it is unsuitable for the catalysis research (Figure R2). In contrast, the sizes of the nanoparticles of Ru and Ir are ~20 nm and ~10 nm, respectively (Figure R3, 4). The XPS spectra show that the negatively charged species are formed on the Ir surface but not on Ru (Figure R5). Compared with Ru/C and Ir/C, the HER performance of the Ru/MgO and Ir/MgO is much better (Figure R6), indicating that an abundance of H^+ created by MgO could also improve the alkaline HER activity of other noble metals. We would like to remark that the HER performance of Ir/MgO is still not as good as that of Pt/MgO, probably due to the inherent difference of Pt and Ir that Pt has the most **approachable zero hydrogen absorption energy**. Besides, the negatively-charged $Pt^{\delta-}$ is also beneficial for the H_3O^+ accumulation, which is evidenced by operando XANES.

Figure R1: XRD patterns of (a) Au/MgO, (b) Ru/MgO and (c) Ir/MgO.

Figure R2. The images of (a) TEM, (b) HRTEM and (c) EDX mapping of Au/MgO.

Figure R3. The images of (a) TEM, (b) HRTEM and (c) EDX mapping of Ru/MgO.

Figure R4. The images of (a) TEM, (b) HRTEM and (c) EDX mapping of the Ir/MgO.

Figure R5. XPS spectra of (a) Ru/MgO and (b) Ir/MgO.

Figure R6. (a) Linear sweep voltammetry (LSV) curves of Pt/MgO, Ru/MgO, Ir/MgO, Pt/C, Ru/C and Ir/C.

Revision,

Accordingly, we have added in the revised manuscript the statement: “*In addition, we synthesized other noble metal (Au, Ru and Ir) nanoparticles supported on MgO to replace Pt (Supplementary Fig. 30-33). Compared with Ru/C and Ir/C, the HER performance of the Ru/MgO and Ir/MgO is much better, indicating that an abundance of H^+ created by MgO could also improve the alkaline HER activity of other noble metals. Structural characterizations show that negatively-charged metal sites are also formed on the Ir surface (Supplementary Fig. 34). Nevertheless, the HER performance of Ir/MgO is still not as good as that of Pt/MgO (Supplementary Fig. 35), probably due to the inherent difference of Pt and Ir that Pt has the most approachable zero hydrogen absorption energy.*”

2)Would it be possible to reduce the amount of oxygen-vacancy in MgO and study its effects on the catalysis of Pt/MgO?

Reply: We sincerely appreciate the reviewer’s insightful suggestion. To explore the relationship between catalytic performance and the concentration of oxygen vacancies, we adopted several methods to control the amount of the oxygen-vacancy in MgO.

First, we tried to tune the annealing temperature to control the amount of oxygen vacancy. However, different annealing temperatures seriously change the size of Pt particles (Figure R7), which makes it not a suitable method to reveal the relationship between the concentration of vacancies and the catalytic performance.

Next, we used a method of Ar plasma irradiation to tune the concentration of oxygen vacancies, based on our previous work to create sulfur vacancies on the MoS₂ by Ar plasma irradiation [ACS Appl. Mater. Interfaces 2019, 11, 31155–31161]. We

transferred the Pt/MgO into a clear cylindrical cavity and treated it by mild Ar plasma irradiation. EPR measurements were conducted to monitor the Ar plasma irradiation effects. As shown in Figure R8a, the amount of oxygen vacancies increases with the irradiation time. Then we tested the HER linear sweep voltammograms for these samples. There was no significant change in the HER property (Figure R8b). This is not surprising because it is the H^* desorption, not the water dissociation, is the rate-limited step. Even at a relatively low concentration of oxygen vacancies, the strong water dissociation ability of V_O -MgO is sufficient to provide a local H^+ abundant environment for Pt. Therefore, further increasing the amount of oxygen vacancies of V_O -MgO has little effect on the HER efficiency.

Finally, as the reviewer suggested, we also tried to reduce the amount of oxygen vacancy in MgO and test the catalysis. For this purpose, we used the diluted acid ($10 \mu\text{M H}_2\text{SO}_4$) etching strategy to gradually reduce the amount of oxygen vacancy in MgO. The EPR and LSV results are shown in Figure R9. With the increase of etching time, the EPR spectra show that the oxygen vacancy decreases in concentration. More importantly, the HER performance of Pt/MgO is also deteriorated with the decreased amount of oxygen vacancies (Figure R9), indicating that the local acid-like environment created by MgO truly plays an important role in the HER process.

Summarizing the above results, there exists a critical amount of oxygen vacancy in MgO, above which the HER performance could not further be improved, while below which the HER performance is deteriorated.

Figure R7: TEM images of Pt/MgO synthesized at different temperatures: (a) 750 °C, (b) 800 °C, (c) 850 °C.

Figure R8: (a) The EPR spectra, (b) Linear sweep voltammetry (LSV) curves of the Pt/MgO samples with different Ar plasma irradiation time.

Figure R9: (a) The EPR spectra, (b) Linear sweep voltammetry (LSV) curves and (c) the corresponding Tafel plots calculated from the LSV curves of the Pt/MgO samples with different diluted acid etching time.

3) Without clear evidence that the acid-like environment coming from MgO is the key for the better catalyst and not just the modification of Pt by substrate effects, the work is not showing a new design route.

Reply: Thanks for the reviewer's critical comment and kind suggestion. To clarify the reviewer's concern regarding the real key factor for improved HER, we carried out additional comparative electrochemical tests and calculations.

Experimentally, to demonstrate that the negatively charged $\text{Pt}^{\delta-}$ alone is not enough for the improved HER catalysis, we firstly replace MgO by another substrate that can induce negative-charged $\text{Pt}^{\delta-}$ but has not strong H_2O dissociation ability. Based on literature reports on the negative-charged $\text{Au}^{\delta-}$ and $\text{Ni}^{\delta-}$ [ACS Catal. 2019, 9, 2707–2717; ACS Catal. 2017, 7, 7600–7609] on TiO_2 , which also possesses the F centers, we used TiO_2 as the alternative support to induce the negatively-charged $\text{Pt}^{\delta-}$. As shown in Figure R10, we successfully obtained the Pt/ TiO_2 . Then we carried out the

Pt 4f XPS measurements for the synthesized Pt/TiO₂ (Figure R 11). As expected, a peak at 70.6 eV is observed in the lower binding energy side than the Pt⁰ state, suggesting the formation of negatively-charged Pt^{δ-} atoms. The LSV tests as shown in Figure R12 indicate the relatively poor HER activity of Pt^{δ-}/TiO₂, with an overpotential of 60 mV at the current density of 10 mA cm⁻². The significantly lower activity of Pt^{δ-}/TiO₂ than Pt^{δ-}/MgO leads us to conclude that the negatively-charged Pt^{δ-} species could not be responsible for all the improvement on HER performance. More importantly, the Tafel slope of Pt^{δ-}/TiO₂ is 62 mV dec⁻¹, suggesting the water dissociation is still the rate determining step of the HER for Pt^{δ-}/TiO₂.

In addition, we used the diluted acid (10 μM H₂SO₄) etching strategy to gradually decrease the amount of oxygen vacancies in Pt/MgO (Figure R13a). The HER property of Pt/MgO becomes worse with the reduced oxygen vacancies (Figure R 13c), indicating the important role of V_O-MgO in catalyzing HER. Moreover, after 30 min of acid-etching treatment, the Tafel slope increases from 37 to 65 mV dec⁻¹ (Figure 13d), indicating the changed rate-determining step of HER from H₂ desorption to the water dissociation. The Pt 4f XPS measurements for the Pt/MgO@10min of acid-etching (Figure R13b) show that the Pt still maintains its negatively charged state. These results further reinforce our claim that the better catalytic activity does not just come from the modification of Pt. The local acid-like environment created by the strong water dissociation ability of MgO and the proton aggregation ability of negative valence of Pt are the keys to the high HER activity.

We also established the theoretical model for calculating the H₂O dissociation energy on the negatively charged Pt sites, and compare it with that of V_O-MgO, as shown in Figure R14. The dissociation energy of H₂O into OH and H on negatively charged Pt site is 0.34 eV, significantly higher than that of V_O-MgO (-1.7 eV). Such a water dissociation ability of negatively charged Pt is similar to Pt/C; however, the electrocatalytic HER performances of our V_O-MgO supported Pt is significantly superior to Pt/C, because the V_O-MgO support solves the problem of poor water dissociation ability of catalysts in alkaline HER and creates a local acid-like environment for the Pt. These results imply the modification of Pt^{δ-} cannot boost such a high alkaline HER activity.

In summary, we have separately studied the effects of negatively-charged Pt^{δ-} and oxygen vacancy-rich MgO on the HER performance from both experimental and computational sides. The results show that the excellent HER activity of Pt/MgO originates from the favorable local acid-like environment created for the active sites of

negative-charged $\text{Pt}^{\delta-}$, rather than only from the electronic modification of Pt.

Figure R10. The images of (a) TEM, (b) HRTEM and (c) EDX mapping of Pt/TiO₂.

Figure R11. (a) XRD and (b) XPS spectra of Pt/TiO₂.

Figure R12. (a) Linear sweep voltammetry (LSV), (b) the corresponding Tafel plots calculated from the LSV curves of Pt/TiO₂ and Pt/MgO.

Figure R13: (a) The EPR spectra, (b) the XPS spectra, (c) Linear sweep voltammetry (LSV) curves, and (d) the corresponding Tafel plots calculated from the LSV curves of the Pt/MgO samples with different with etching time.

Figure R14: The water dissociation energy of V_o -MgO and negatively charged Pt.

Revision:

Accordingly, we have added in the revised manuscript the statements: *“More importantly, a series of experiments and simulations, including the synthesis of negatively Pt nanoparticles supported on TiO₂ (without strong water dissociation ability), reducing the amount of oxygen vacancy in MgO, and the calculations of the water dissociation ability of negative-valence Pt (Supplementary Fig. 25-29), show that the high HER activity comes from the synergistic effect of local reaction environment and the negatively charged Pt^{δ-}, rather than only comes from the modified electronic structure of Pt.”*

4)The DFT calculations included by the author were performed using standard methodologies. However, details are missing about how the results presented in Fig. 5a to 5c were acquired; how was the atomic charge difference calculated, is the difference between what? The authors mention Pt nanoparticles in the test, but Fig 5a to 5c looks like an overlayer of Pt over MgO, and no Pt nanoparticles on MgO; is the Pt overlayer on MgO strained? The authors included calculations with a cluster of Pt⁵ in Fig 5d. Charge analysis for such cluster on MgO systems will be more relevant here, even though the experimentally prepared nanoparticles have sizes of 5 nm; a cluster of Pt₅ has less than 0.5 nm diameter and quantum confinement effects are present.

Reply: Before the point-by-point reply to the questions on the DFT simulations, we would like to detailly describe the models used for the theoretical investigations in this work.

We have used two models for different purposes. (1) For the simulations of the dissociation of water on the surface of MgO and the following migration of proton towards Pt nanoparticles, the model should contain MgO surface, O vacancies and a Pt cluster. Hence a “MgO surface-O vacancy-Pt₅” model was built for these simulations. We have to use a much smaller Pt₅ cluster to represent the Pt nanoparticle of 5 nm because a larger Pt cluster would cause much larger size of model which is beyond our computational resources. Actually, the model with the Pt₅ cluster on MgO already contains 199 atoms. Besides, this simulation focuses on the production and migration of protons. Since the small Pt cluster’s affinity for proton is close to that in the large Pt nanoparticles, this approximation should be acceptable. (2) Another model is the “MgO[001]-Pt[100] interface” model, which is used to study the interfacial charge transfer between MgO and Pt nanoparticles. As you commented, the quantum confinement could dramatically affect the position and occupation of energy levels of

small clusters, including the Fermi level, which is crucial for charge transfer. Hence the layered slab models are widely adopted for interfacial charge transfer simulations instead of the cluster model.

For the results presented in Fig. 5a to 5c, first we apologize for the typo in the caption of Fig. 5 in the previous version. The correct caption for (a-c) should be “**Atomic charges of Pt (a), Pt-MgO (b), Pt-MgO-Vo (c).**” The atomic charges displayed in Fig. 5a-c are Bader charges based on the Atoms-in-Molecules (AIM) analysis (Bader 1985) calculated with an accelerated algorithm developed by G. Henkelman and co-workers (Tang, Sanville et al. 2009). The Bader charge analysis has been widely used to theoretically measure the interfacial charge transfer.

For the overlayer model for the atomic charge calculation, we chose the aforementioned model “MgO[001]-Pt[100] interface”, instead of the “MgO surface-O vacancy-Pt₅” model to avoid the quantum confinement effect on charge transfer. A slight strain of about 2% [from $a=3.97$ Å (Fig. 5a) to $a=3.80$ Å (Fig. 5b-c)] exists in the MgO[001]-Pt[100] interface model due to the lattice mismatch. But the comparison between Fig. 5a-c reveals that the atomic change of the Pt layer is mainly from electron transfer from V_O to Pt, instead of the effect of strain.

Revision,

Accordingly, the details of the models used for the theoretical investigations were added in the section of “DFT Calculation Details” in the supporting information.

Reviewer #2 (Remarks to the Author):

The authors have synthesized a Pt/MgO catalyst using a MOF-derived MgO support for alkaline HER. The authors claim that the as-prepared Pt/MgO generates a local acid-like environment in the alkaline medium providing enhanced activity. They further show that vacancy-rich MgO nanosheets initiate the water dissociation and produce H₃O⁺, which finally accumulate around Pt^{δ-} to evolve H₂. Although the concept is not novel, it is important to note that the authors have used several advanced operando techniques to carry out this work. Nonetheless, there are some major inconsistencies (see below) in the manuscript that should be addressed before it can be considered for publication in a highly reputed journal like Nature Communications.

Reply: We thank the reviewer for reviewing our manuscript, and we appreciate the

reviewer for these precious comments and suggestions. First of all, about the inconsistencies you mentioned in the comments, we are sorry that we failed to provide a detailed explanation about the *operando* Raman and SR-FTIR characterizations. After carefully considering your questions, we found that the inconsistencies may come from the differences in the experimental details and physical principles in *operando* Raman and SR-FTIR tests. The corresponding explanations are shown in the following.

In addition, we are sorry that we did not clearly express the novelties of this work. It is well known that the HER conversion efficiency in alkaline condition is two or three orders of magnitude lower than in acidic solution. To improve the alkaline HER activity, enormous efforts such as doping, defect engineering and strain engineering have been made, aiming to adjust the electronic structures of the activity sites and thus to tune the binding strength of the relevant molecules. However, there is still a large gap between the activities in the alkaline environment as compared to those in the acidic environment because these strategies could not fundamentally change the reaction barrier in the alkaline condition, making the alkaline HER still sluggish. Recently, researchers come to recognize the important role of local reaction environment in electrocatalysis. However, the preparation of desired reaction environment in the adverse electrolyte requires to deliberately tune the solid/liquid electrochemical interfaces. This task is challenged by the lack of facile and practical strategies to engineer the local reaction environment, as well as by the difficulty in identifying the weak signals arising from interfacial structures in complex reaction conditions. In this work, we study the mechanism of modifying the local reaction environment. Comparing to previous studies, the characteristics and novelties of this work are highlighted as below:

1) In order to deliberately create a desired local acid-like environment, for the first time, we propose a practical pathway by using the oxygen vacancy-rich MgO to generate an abundance of H_3O^+ groups and **using negatively charged $\text{Pt}^{\delta-}$** to accumulate them;

2) The Pt/MgO exhibits an excellent catalytic activity with a very low overpotential of **39 mV at 10 mA cm^{-2}** , much better than 62 mV for 20 wt% Pt/C and **close to the acidic HER behavior of 20 wt% Pt/C (33 mV)**;

3) The proposed mechanism and related intermediates were detected and studied by the **in-situ Raman, synchrotron radiation Fourier transformed infrared spectroscopy (SR-FTIR) and X-ray absorption near-edge structure spectroscopy (XANES)**, all of which are consistent with the computational simulations.

1. First of all, why do the authors see a graphite peak in Figure 3a (Raman spectra) if they recorded the spectra only over Pt/MgO? Moreover, the experimental section also does not provide any details on the carbon contamination or the percentage of carbon left in Pt/MgO. I assume that the G band might be originating from (unremoved) carbon. If it is true, then this peak should show a broader nature at reduced potentials as the amount of adsorbed water will be inflated with potential. However, the figure shows quite the opposite (although the authors claim that it is unchanged).

Reply: We thank the reviewer for pointing out the unclear description of this issue. First of all, we are sorry that we failed to provide the detailed explanation about the Raman measurement in the main manuscript. As you suggested, the G band originates from the (unremoved) carbon because the Pt/MgO sample was derived from the MOFs. However, in the electrolyte, the G-band peak of the graphite ($\sim 1590\text{ cm}^{-1}$) and the H_2O peak ($\sim 1600\text{ cm}^{-1}$) are overlapped in the Raman spectra. Previous studies suggest that the original G-band of the carbon materials ($\sim 1590\text{ cm}^{-1}$) becomes broader because a large amount of adsorbed water increases the shoulder peak of water [J. Electroanal. Chem. 415, 1996, 175-178]. However, in our work, as the reviewer commented, the overlapping peak becomes narrow (what we stated in the text is that the peak intensity of the graphite has not changed). This is due to the weakening of the peak of adsorbed water, which leads to the narrowing of the whole overlapping peak (Figure R15). The reason for the decreased peak intensity of water is the strong water dissociation on the catalyst's surface during the HER process.

Figure R15. In situ Raman spectra of Pt/MgO. Two stretching modes are fitted in light blue and red, respectively.

Revision,

Accordingly, in the original manuscript, the statements “*whereas the peak of H₂O (1600 cm⁻¹) becomes weaker and the G-band of graphite (1580 cm⁻¹) remains unchanged, indicating generating of H₃O⁺ intermediate.*” have been changed to “*whereas the peak of H₂O (1600 cm⁻¹) becomes weaker and the G-band of graphite (1580 cm⁻¹) remains unchanged, indicating the facilitated water dissociation on the surface of MgO and thereby generation of abundant H₃O⁺ intermediate.*”

2. It is quite a strange observation that MgO is responding for H₃O⁺ (Supplementary Fig 7) although the proton adsorption ability of MgO is not adequate as compared to Pt. This is the reason the current density value for MgO is significantly low. Please incorporate additional evidence to prove this point.

Reply: Thanks for the reviewer’s insightful comment. Yes, operando FT-IR spectra in Supplementary Fig. 7 (Supplementary Fig. 8 in the revised manuscript) shows that MgO is responding for H₃O⁺, but this response is not caused by H₃O⁺ adsorbed on MgO, because of the weak proton adsorption ability of MgO as you commented. We are sorry for our unclear description that may have misled you.

One of the key roles of MgO is to generate an abundance of H₃O⁺ intermediates in the double layer, due to its good water dissociation ability. Therefore, it is the free H₃O⁺ in the double layer, rather than the weakly adsorbed H₃O⁺ on MgO, that contributes to IR signal at ~3525 cm⁻¹ (Science 2003, 299, 1375; Science 2005, 308, 1765). Previous reports show that the signal intensity of the in-situ infrared or Raman spectra **are mainly dependent on the steady-state concentration of intermediates** [Catalysis Today, 1999, 49, 467-484; Journal of Catalysis, 2001, 203, 104-121]. For the step of generation of H₃O⁺ (it is not the rate determining step for the whole HER process in this work), as the potential decreases, the reaction degree of this step becomes stronger, more intermediates are generated and accumulated during the reaction. Thus, the detected IR signal at ~3525 cm⁻¹ corresponding to the H₃O⁺ species becomes stronger. And the current density value for MgO is low because MgO only possesses good water dissociation but lacks the hydrogen ad-desorption ability; therefore the whole HER efficiency is hindered by the subsequent generation of H₂ via either the Heyrovsky or the Tafel step.

3. If it is assumed that in Pt/C composite Pt is partially positively charged, then why do

the authors observe H_3O^+ signal in SR-FTIR (Supplementary Fig 9).? Also, the pH effect on the local acidic medium is not logical. With pH, the adsorption of proton increases linearly. The signal intensity for H_3O^+ is higher (if it is not a spectral artifact) in pH 12 as compared to pH 13. In contrast, the Raman spectra did not give any signal for H_3O^+ (Supplementary Fig 7). Besides, this claim does not correlate with the study done in Nat. Commun. 10, 4876, 2019.

Reply: We sincerely appreciate the reviewer's professional comment. First of all, we would like to apologize for the mistake we made in Supplementary Fig 9. As we stated in the main manuscript: "*It is our aim in this work to engineer this unique H_3O^+ species for creating a desired local acidity under widespread alkaline conditions (Supplementary Fig. 9)*", the supplementary Fig. 9 (Supplementary Fig. 10 in the revised manuscript) is actually for the sample of Pt/MgO rather than for Pt/C. It indicates that Pt/MgO could create a local acid-like environment in a wide pH range. We are sorry for the typo.

Synchrotron radiation (SR) techniques have proved to be advantageous in catalysis studies in that they can give sufficient signals of intermediates at the liquid-solid interface [Nat. Energy 2019, 4, 115–122; Nat. Chem. 2020, 12, 717–724]. The schematic diagram of the synchrotron radiation-based in situ reflectance infrared Fourier transform spectroscopy is shown in Figure R16. The signal intensity is affected by many factors, such as the infrared absorption of liquid film, concentration of intermediate, light intensity decays and so on. Practically, it is hard to ensure that the samples are uniformly distributed on the electrode. Therefore, every time when we change the sample or test point, the infrared signal will be influenced by these factors. Besides, the IR spectroscopy is only sensitive to the functional groups within ~3 molecular monolayers [Nano Energy 2020, 77, 105121]. As commented by the reviewer, ideally, pH will affect the structure of the double-layer at the solid-liquid interface. However, in the actual test process, the test point is a local area of the electrode surface. The complexity of numerous external factors makes it actually impossible to have a linear relationship between the signal intensity and the pH value. To sum up, the peak intensity of operando SR-FTIR spectroscopy could not be used for quantitative analysis, but suitable for qualitative analysis such as for the detection of intermediates [Catalysis Today 2017, 283, 176–184]. This is the reason why the H_3O^+ signal intensity is considerably higher in pH 12 than in pH 13, but not higher by 10 times.

Similar to the infrared spectrum, Raman spectrum is also the molecular vibration spectrum and can reflect the characteristic structure of intermediate

molecules. However, the ordinary Raman scattering is a very weak process, where the light intensity is only 10^{-10} of the incident light intensity. Thus, the Raman signal without surface enhanced Raman scattering (SERS) effect is much lower than the synchrotron radiation infrared spectrum. Previous studies have shown that some metals such as Au, Ag, Cu, Pt yield the surface enhanced Raman scattering (SERS) effect [J. Phys. Chem. B 2006, 110, 4, 1837–1842]. This is the reason why we did not observe the Raman signal of H_3O^+ on pure MgO, but did on Pt/MgO. In order to confirm that H_3O^+ is indeed generated on MgO, we loaded Au nanoparticles on the surface of pure MgO and then did the Raman test. The results are shown in below (Figure R17). As we expected, an obvious H_3O^+ peak is observed, which confirms that MgO is beneficial for water dissociation and generates an abundance of H_3O^+ group.

Lastly, in the article mentioned by the reviewer [Nat. Commun. 10, 4876, 2019.], the authors also observed a local acidic environment in a very specific pH (pH=13). As we have stated above, because the Supplementary Fig. 9 is for the sample of Pt/MgO rather than for Pt/C, our work is not in conflict with them.

Figure R16: Schematic diagram of the synchrotron radiation-based in situ reflectance infrared Fourier transform spectroscopy.

Figure R17: The operando Raman spectra of Au_MgO in 1M KOH.

4. In theoretical calculation, the calculated energy for adsorbed proton (H^*) on Pt^0 is not close to its well-known reported value. (see Chem. Sci., 2019, 10, 9165-9181).

Also, do the authors have any evidence for hydroxyl-water-cation adduct?

Reply: The free energy for the HER in Chem. Sci., 2019, 10, 9165-9181 was calculated for the Pt [111] surface, while the free energy in our work was calculated for the Pt [100] surface. Previous studies have indicated that the hydrogen adsorption energy on the Pt[100] is ~ 0.2 eV lower than that on Pt [111] (Adsorption Energetics of Atoms and Diatomic Gases with Electrocatalysis Approach towards Hydrogen and Oxygen Evolution Reaction on Pt Surfaces), which can explain why our free energy is lower than the value in Chem. Sci., 2019, 10, 9165-9181. We chose the Pt[100] for HER free energy calculation because we intended to study the effect of substrate-Pt charge transfer on HER and the Pt[100] has the least lattice mismatch with MgO substrate.

The existence of hydroxyl-water-cation adduct is well documented in previous works [Nat Chem 2009, 1, 466–472, Science 2011, 334, 6060, 1256-1260; J. Am. Chem. Soc. 2019, 141, 3232–3239]. First of all, we used the deuterium (2H , D) nuclear magnetic resonance (NMR) to detect the water-cation adduct. Previous works have shown that the methanol inserted into the inner K^+ solvation sheaths and affects the K^+ solvation balance by interacting with coordinated water [Angew. Chem. 2021, 133, 7442-7451]. As shown in Figure R18a, with the introduction of methanol, the 2H peak shifts to the higher side, indicating that K^+ exhibits a stable double-layer solvated structure $(H_2O)_x-AM^+$ in KOH electrolyte. Next, according to the hard-soft acid-base (HSAB) mechanism, as the Lewis acid hardness increases in the order $Li^+ < Na^+ < K^+$, the interaction energy between OH_{ad} and $(H_2O)_x-AM^+$ within $OH_{ad}-(H_2O)_x-AM^+$ decreases.

To trace the interaction between OH_{ad} and $(\text{H}_2\text{O})_x\text{-AM}^+$, we added LiClO_4 (K^+ was not chosen because of the low solubility of KClO_4) into the 1M KOH electrolyte and conducted the CO stripping experiment. As shown in Figure R18c, the CO oxidation potential shifts to the higher potential after adding the Li ions, because the increased concentration of $(\text{H}_2\text{O})_x\text{-Li}^+$ increases the interaction between $(\text{H}_2\text{O})_x\text{-AM}^+$ and OH^* and meanwhile promotes the desorption of OH_{ad} (Figure R18d), in line with the previous results [Energy Environ. Sci., 2020, 13, 3064-3074]. These results indicate the existence of the hydroxyl-water-cation adduct.

Figure R18: (a) ^2H NMR spectra, (b) Schematic of changes in the K^+ -water adduct, together with methanol addition. (c) CO stripping (inset) of Pt in 1 M KOH and plus LiClO_4 . (d) Schematic illustration of HER and CO oxidation mechanism.

Revision:

we have added in the revised manuscript the statement: “*the existence of $\text{OH}_{\text{ad}}\text{-(H}_2\text{O)}_x\text{-K}^+$ is proved by the deuterium (^2H , D) nuclear magnetic resonance (NMR) CO stripping experiments (Supplementary Fig. 41)*”.

5. The XPS shift for Mg after inclusion of Pt looks empirical (supplementary Fig. 13). The figure reads more than a 0.7 eV shift which is probably uncorrected from the standard carbon profile. In addition, the authors should note that the comparison of Pt

4f spectra between Pt foil and Pt/MgO may not provide conclusive data (Fig. 3d) because the size of the Pt in the prepared catalyst is sufficiently small even though it belongs to the bulk Pt. Ideally, Pt nanoparticles with similar sizes should be used for comparison.

Reply: We truly appreciate the reviewer for reminding us these issues. The XPS spectra in this work were recorded on an ESCALAB MKII with Mg K α ($h\nu = 1253.6$ eV) as the excitation source. We have carefully checked the original data. The survey XPS and C 1s XPS spectra are given below. The C 1s XPS (Figure R19b) shows that the binding energy of the Pt/MgO and MgO samples were corrected to 284.5 eV. Besides, we are sorry that we did not label the fine values of the main peaks of the two samples (supplementary Fig. 13, which is supplementary Fig. 14 in the revised manuscript). The energy shift in supplementary Fig. 13 seems big because the range of the X-axis is small (from -44 to 56 eV). In fact, the peak shift is only 0.48 eV, as seen from the labeled line values of the main peaks for Mg, which are 51.00 and 50.52 eV, respectively, for Pt/MgO and MgO (Figure R20). In order to further validate our data testing or processing, we recollected the XPS spectra of these two samples and calibrated the binding energy by adding Au powder as the internal standard (Figure R21). The results show that the peak shift of Mg 2p XPS is 0.42 eV, very close to the value of 0.48 eV by using the carbon standard. Similar magnitude of energy shift on MgO has also been observed by other researchers in the strong electronic interaction effect system. For instant, M.S. Mastuli *et. al.* show a shift of 0.42 eV in BE values of Mg 2p in NiO/MgO and ZnO/MgO [International Journal of Hydrogen Energy 2017, 42, 11215-11228].

In addition, for the Pt 4f XPS of Pt/MgO, we found a shoulder peak at the low energy end of the zero-valence peak. As suggested, we used the Pt 4f XPS of Pt nanoparticle of ~5 nm to replace the Pt foil XPS, as shown in Figure R22. Compared with Pt foil, the position of the Pt 4f XPS of Pt nanoparticles shows a slight shift. The shoulder peak of Pt/MgO is still located at the lower-energy side than the peak position of Pt nanoparticles, confirming again the existence of Pt $^{\delta}$.

Figure R19: (a) The survey XPS spectra and (b) C 1s XPS spectra of Pt/C and Pt/MgO.

Figure R20: Mg 2p XPS spectra of MgO and Pt/MgO.

Figure R21: (a) Au 4f XPS spectra of Au/MgO and AuPt/MgO. (b) Mg 2p XPS spectra of MgO and Pt/MgO where Au powder was added as the internal standard.

Figure R22: XPS spectra of Figure 3d by using the spectrum of Pt nanoparticles of 5 nm to replace the spectrum of Pt foil.

6. What do the authors mean by “after reaction” in supplementary Fig 8? How is it correlated with reversibility?

Reply: Supplementary Fig. 8 is the operando infrared measurement after catalysis reactions. It turned out that the potential-dependent signal of H_3O^+ is reversible, manifesting that the local acid-like environment is formed during the process of HER rather than a spectral artifact.

7. What was the main reason to use MOF for the synthesis? Is it because of high surface area or to generate vacancies?

Reply: Thank you for your comment. We chose to use MOF for the synthesis based on the two reasons: First, as you mentioned, it is easy to generate an abundance of vacancies during the self-pyrolysis process of MOFs [Nat. Rev. Mater. 2018, 3, 17075]. As shown in Supplementary Fig. 13, the g value of 2.00 corresponds to the oxygen vacancies formed by the oxygen evaporation at high-temperature. Second, in order to synthesize the Pt-MgO system with strong electronic metal-support interaction (EMSI) effect, we immersed Pt ions into the pores of Mg-MOF. The spatial confinement effect during MOFs pyrolysis contributes to the recombination and dispersion of Pt nanoparticles, resulting in a composite interface with strong electronic metal-support interaction.

Revision,

Accordingly, in the original manuscript, the statements “*In order to synthesize Pt/MgO catalyst with strong electronic metal-support interaction (EMSI) effect*” have been changed to “*In order to synthesize Pt/MgO catalyst with strong electronic metal-*

support interaction (EMSI) and an abundance of oxygen vacancies,” in the reserved manuscript.

8. MgO could form Mg(OH)₂ in KOH solution (at least at the surface). An X-ray powder diffraction of the catalysts before and after the reaction would help to get more insights.

Reply: According to the suggestion of the reviewer, we collected the powder XRD pattern for the Pt/MgO catalyst after 20 h long-time of HER test in the 1M KOH solution, as shown in Figure R23. The XRD patterns before and after the HER reaction keep almost identical. No diffraction peak related to Mg(OH)₂ is observed in the XRD pattern of the used catalyst. This suggests the stability of MgO during the HER test.

Figure R23: The X-ray powder diffraction of the Pt/MgO before and after 20 h of HER reaction in the 1M KOH solution.

Revision,

Accordingly, in the original manuscript, the statements “*the TEM images (Supplementary Fig.16) of Pt/MgO after continuous operation also keep the original structure*” have been changed to “*the TEM images (Supplementary Fig.17) and XRD spectra (Supplementary Fig.18) of Pt/MgO after continuous operation also keep the original structure.*” in the revised manuscript.

9. What is the pH value used for overpotential calculation? and how this pH value was

measured?

Reply: The pH=0 was used for the DFT overpotential calculation as used in many previous studies [Trends in the Exchange Current for Hydrogen Evolution]. The pH value provides only a constant term for HER free energy, hence it does not affect the trend of the HER overpotential between different catalysts. Experimentally, the pH value was measured by a benchtop pH meter (PHS-3G, Shanghai INESA Scientific Instrument Co., Ltd.). The error range of this pH instrument is ± 0.01 .

10. How does the electrochemical surface area influence the catalytic activity of Pt/C and Pt/MgO? Similarly, the Faradaic efficiency of the reaction should be measured.

Reply: These are good questions and suggestions. According to your suggestion, we calculated the electrochemically active surface area (ECSA) of the Pt/C and Pt/MgO and present their ECSA-normalized HER polarization curves. ECSA of the Pt/C and Pt/MgO were determined from its CV curve in 1 M KOH electrolyte at the scan rate of 50 mV s^{-1} . As shown in Figure R24, the amounts of charge exchanged during the electro-adsorption (Q') and desorption (Q'') of H_2 on Pt can be calculated using the following equation:

$$Q = \frac{1}{\nu} \int_{E_1}^{E_2} IdE,$$

where ν is the scan rate. The region contributed by the capacitive current from the double layer capacitance is deducted from the total charge. The coulombic charge of H_2 desorption (Q_H) on Pt catalysts can be calculated from the equation:

$$Q_H = \frac{1}{2} (Q' + Q'').$$

The ECSAs of Pt/C and Pt/MgO are calculated by the equation:

$$\text{ECSA} = \frac{Q_H}{0.21},$$

where the constant ($0.21 \text{ mC}\cdot\text{cm}^{-2}$) represents the charge required to oxidize a monolayer of H_2 on Pt. The charges Q_H , Q' , Q'' and ECSAs of Pt/C and Pt/MgO are summarized in Table R1.

The ECSA-normalized HER polarization curves of the Pt/C and Pt/MgO are shown in Figure R24c. Evidently, Pt/MgO still has apparently larger current density than Pt/C, indicating the much higher intrinsic catalysis activity in Pt/MgO. Considering that the ECSA in Pt/MgO is lower than Pt/C, we conclude that the enhanced activity of HER in Pt/MgO is more possibly ascribed to the excellent intrinsic activity rather than to the increased ECSA.

In addition, The Faradaic efficiency (FE) was calculated by comparing the

measured amount of H₂ generated by cathodal electrolysis with the calculated amount of H₂ (assuming an FE of 100%). As shown in Figure R25, the amount of produced H₂ was basically consistent with the calculated production, indicating the nearly 100% FE for the HER in base.

Figure R24. ECSA calculation of (a) Pt/C, (b) Pt/MgO in 1.0 M KOH. (c) ECSA-normalized HER polarization curves.

Catalyst	Q' (mC)	Q'' (mC)	Q_H (mC)	ECSA (cm ²)
Pt/C	0.84	0.96	0.9	4.20
Pt/MgO	0.2	0.5	0.35	1.67

Table R1. Q_H , Q' , Q'' and ECSA of Pt/C and Pt/MgO.

Figure R25: The amount of gas theoretically calculated and experimentally measured versus time over HER for (a) Pt/MgO and (b) Pt/C, Faradic efficiency of hydrogen production for (c) Pt/MgO and (d) Pt/C.

Revision:

Accordingly, in the original manuscript, the statements “Moreover, the ECSA-normalized HER polarization curves (Supplementary Fig. 22) and the turnover frequency (TOF) (Supplementary Fig. 23) suggest that the improvement of HER catalytic performance is mainly due to the increase of catalysis activity of the intrinsic active sites. Reducing the catalysts loading on the cathodes of Pt/C and Pt/MgO has no impact on the interaction between H^* and the catalyst surfaces (Supplementary Fig. 23), suggesting the excellent intrinsic HER activity of Pt/MgO.” and “In addition, the Pt/MgO shows nearly 100% Faradaic efficiency in base (Supplementary Fig. 24)” have been added in the revised manuscript.

11. Authors should simulate the EXAFS spectra.

Reply: We deeply appreciate the reviewer’s suggestion. According to the reviewer’s comment, we simulated the Pt L_3 -edge EXAFS spectra and the result is shown below. The obtained coordination number of the first shell of Pt is 10.1, which is significantly lower than the value (12) of Pt foil. The decreased coordination number is mainly a result of the size effect, because there is a significant amount of unsaturated coordination bonds on the surface of nano-sized materials.

Figure R26: k^2 -weight FT-EXAFS fitting curves (a) and corresponding $\text{Re}(k^2\chi(k))$ oscillations (b) of Pt/MgO at Pt K-edge.

Sample	Path	R (Å)	N	σ^2 (10^{-3} Å ²)	ΔE_0 (eV)
--------	------	---------	-----	---	-------------------

Pt/MgO	Pt-Pt	2.75 ± 0.01	10.1 ± 0.9	4.9 ± 0.8	6.7 ± 1.3
--------	-------	-----------------	----------------	---------------	---------------

Table R2. Structural parameters of Pt/MgO obtained from EXAFS curve-fitting.

Revision,

Accordingly, in the revised manuscript, we have added the statements: *“The EXAFS fitting results (Supplementary Fig.6 and Table 1) show that the coordination number of the first shell of Pt is 10.1, significantly lower than the value (12) of Pt foil. The decreased coordination number is mainly due to the existence of a significant amount of unsaturated coordination bonds on the surface of nano-sized materials.”*

Reviewer #3 (Remarks to the Author):

The authors demonstrate a methodology to tune the local reaction environment such that a local acid like environment is created in an alkaline medium which results in a catalyst surface with a superior HER performance. The catalyst synthesized is promising with an overpotential of 39 mV at 10 mA/cm² close to acidic HER activity of Pt/C which has an overpotential of 33 mV at a similar current density. Although the data presented is promising, the manuscript lacks in certain areas particularly in providing an explanation on some of the controversy surrounding alkaline HER reaction. I do think this work is significant to the field but some clarifications are necessary. There should be minor revisions based on the following questions and comments before consideration for publication.

Reply: We thank the reviewer for spending valuable time on reviewing our manuscript and giving positive evaluation to our work. All the comments and suggestions from the reviewer are greatly appreciated. As the reviewer commented, the mechanism of the alkaline HER is still unclear. Despite several theories proposed, such as hydrogen binding energy (HBE) theory, water dissociation theory and interface water and/or anion transfer theory, the reaction mechanism of the alkaline HER in the real reaction environment is still under debate. For example, Markovic et al. proposed that the extra water dissociation caused the slow alkaline HER kinetics [Angew Chem Int Ed 2012, 51, 12495–12498; Nat Mater 2012, 11, 550–557]. However, Yan et al. [Nat. Commun. 2015, 6, 5848; J. Electrochem. Soc. 2018, 165, 27–29] demonstrated that the hydrogen underpotential adsorption/desorption peak positions were strongly dependent on the

HBE. They proposed that the HBE could be the sole descriptor of the HER kinetics. However, the HBE theory can hardly explain the phenomenon that the position of H_{upd} peak does not change too much but the HER activity largely alters under different pH conditions on the Pt (111) surface [Nat. Energy 2017, 2, 17031]. Recently, Jia et al. proposed that the process of removal of OH^* from the electric double layer to the bulk solution is the RDS for alkaline HER [J. Am. Chem. Soc. 2019, 147, 3232], and the kinetics of OH^* transfer can be altered by changing the metal cation (AM^+). Although tremendous efforts have been made, it is still unsure which one is the key factor in boosting the alkaline HER kinetics. Even for the most prominent catalyst (Pt), the HER conversion efficiency in alkaline condition is two or three orders of magnitude lower than in acidic solution. To improve the alkaline HER activity, enormous efforts such as doping, defect engineering and strain engineering have been made, aiming to adjust the electronic structure of the activity site thus to tune the binding strength of the relevant molecules. However, there is still a large gap in activity compared to that of acidic environment because these strategies could not fundamentally change the sluggish reaction barrier in alkaline condition. Therefore, one of the most reasonable strategies is to create a local acidic environment for the Pt sites. In this work, we selected Pt/MgO as the prototypical example, by virtue of multiple physicochemical interactions between the oxygen vacancy-rich MgO and Pt nanoparticle, both of which play important roles in water dissociation, H_3O^+ migration and H^* desorption. In addition, according to the reviewer's comments and suggestions, we have also carefully analyzed and discussed our data for a better presentation of our findings. New results and discussions have been added to our revised manuscript. Point-to-point responses are listed below.

Revision:

Accordingly, in the original manuscript, the originated statements “*The sluggish reaction rate of the alkaline HER is due to the slow step of H_2O dissociation that provides protons for the subsequent reactions,¹⁹ but this step is not necessary in acidic solution.*” has been changed to “*The alkaline HER kinetics is still elusive and several schools of thought on the slow kinetics of the alkaline HER have been proposed, including hydrogen binding energy (HBE) theory, water dissociation theory and interface water and/or anion transfer theory. Nevertheless, it is a general consensus that during the alkaline HER the sluggish Volmer step directly or indirectly impact the rate determining step, but this step is unnecessary in acidic solution.*”

1) I do not understand the purpose of Supplementary Fig 9, it shows H_3O^+ . What is the difference between Supp Fig 9 and Supp Fig 7d? What is different in how the measurement is taken?

Reply: We sincerely appreciate the reviewer's insightful comment. First, we would like to apologize for the mistake we made in Supplementary Fig 9 (Supplementary Fig. 10 in the revised manuscript) in the original manuscript: it is actually for the sample of Pt/MgO rather than for the sample of Pt/C. We are sorry for this typo. This can be inferred from the sentence in the main manuscript: "*It is our aim in this work to engineer this unique H_3O^+ species for creating a desired local acidity under widespread alkaline conditions (Supplementary Fig.9).*"

In Ref. 24 [*Nat. Commun.* 2019, **10**, 4876], Wang et al. discovered the H_3O^+ species on the surface of Pt nanoparticles on the commercial Pt/C catalysis under a specific condition (pH=13). Inspired by this finding, in our work, we aim to seek a suitable system capable of creating and engineering the desired local acidity under widespread alkaline conditions, as well as to elucidate the underlying mechanism. Then we measured the operando SR-FTIR spectra of the Pt/MgO catalyst at different pH (**Supplementary Fig. 9**). Supplementary Fig. 9 (Supplementary Fig. 10 in the revised manuscript) shows that our system could create a local acid-like environment in a wide pH condition. In the original Supplementary Fig. 7d (Supplementary Fig. 8 in the revised manuscript), samples of MgO and Pt/MgO were compared at pH=14 with operando XANES spectra. These joint characterizations indicate that abundant H_3O^+ has been generated on the surface of MgO and forms a local H_3O^+ enriched, acid-like reaction environment around Pt nanoparticles in Pt/MgO. The difference of the measurement between Supplementary Fig. 9 (Supplementary Fig. 10 in the revised manuscript) and Supplementary Fig. 7d (Supplementary Fig. 8d in the revised manuscript) is caused by the different pH condition.

2) Incorrect labelling of Fig. 6 (labelled as Figure 5)

Reply: We thank the reviewer for pointing out this mistake. Now the incorrect labelling has been corrected as Fig. 6 in the revised manuscript. And we have also carefully checked the entire manuscript for grammatical and formatting errors.

3) Can the authors comment on the OH binding strength of their catalyst? The local reaction environment may not be the only thing that is altered and there might be other

factors in play as well. There was a recent study by Marc Koper that talks about the role of adsorbed hydroxide in HER. There are other studies as well that discuss the role of adsorbed hydroxide. Because of the extensive discussion on adsorbed hydroxide in literature, I feel that it is important to discuss how hydroxide binding strength is changing with MgO compared to Pt/C and if it has any role.

Link to articles:

<https://www.nature.com/articles/s41560-020-00710-8> (The role of adsorbed hydroxide in hydrogen evolution reaction kinetics on modified platinum)

<https://pubs.acs.org/doi/abs/10.1021/acscatal.7b02787> (Adsorbed Hydroxide Does Not Participate in the Volmer Step of Alkaline Hydrogen Electrocatalysis)

Reply: We sincerely appreciate the reviewer's insightful comment. There are extensive debates on the role of the adsorbed hydroxide in the alkaline HER. One view from the aspect of thermodynamics is that the Volmer step is the rate determining step in base and the $\text{OH}_{\text{ad}}\text{-M}$ interaction can serve as a primary descriptor of alkaline HER because an optimized host of hydroxyl generated from water dissociation could facilitate alkaline HER process [Nat Mater 2012, 11:550–557; Nano Energy 2016, 29, 29–36]. Another refuted view is that OH does not directly participate in HER but leads to a decrease in the number of active sites for H adsorption and the thermodynamic explanation of OH is not accurate [ACS Catal. 2017, 7, 8314–8319., J. Electrochem. Soc. 165, 2018, 3209–3221,]. Later, Koper et. al. proposed a complete alkaline hydrogen evolution 3D volcano plot for catalyst design (Figure R27), in which hydrogen binding energy and hydroxide binding energy are used as axis and qualitatively captures trends in alkaline HER kinetics [Nat. Energy 5, 2020, 891–899]. By using experiments and modelled DFT, they showed that the hydrogen binding strength and hydroxide binding strength must both be optimized to improve the catalyst activity.

Based on the Brønsted-Evans-Polanyi-type principles, a catalyst with appropriate OH-binding energy that is neither too strong nor too weak is expected. Previous studies have suggested that CO stripping of Pt-based materials can be used for measuring OH^* adsorption energetics, which is crucial to water dissociation. The CV curves in Supplementary Fig. 38 show that the stripping peak for CO_{ad} oxidation on Pt/MgO has a more negative value than that on Pt/C, indicating a higher OH^* adsorption ability of Pt/MgO over Pt/C. Then we conclude that Pt/MgO significantly accelerates water dissociation [Nat. Commun. 2019, 10, 4876]. And the the Volmer step is not the rate determining step for Pt/MgO. This also agrees with the kinetic analysis of a Tafel slope

of 39 mV/dec in alkaline solution, which indicates that the H_2 recombination is the rate determining step for the overall HER. As a result, the Pt/MgO catalyst shows an optimal H^*/OH^* adsorption, which occurs in the areas of H_2 recombination. In short, the Pt/MgO catalyst creates a local acid-like environment in alkaline solution, which provides Pt with a favorable reaction environment for HER in alkaline condition.

Figure R27: 3D volcano plot for the alkaline HER activity. Copyright 2020, Nature Publishing Group.

Revision,

In the revised manuscript, we have added the statement: *“This is demonstrated by the carbon monoxide (CO) stripping tests (Supplementary Fig. 37). The CV curves show that the stripping peak for CO_{ad} oxidation of Pt/MgO has a more negative value than that for Pt/C, indicating a better water dissociation ability for Pt/MgO over Pt/C.”*

4) The authors argue that the negatively charged Pt creates a local acid-like reaction environment. There are many studies in literature that talk about the potential of zero free charge (pzfc) of Pt in acid vs base. Most of the studies point out that the surface is more negative in base because the pzfc shifts towards more positive potentials. Recently a study by a group at MIT tracked electric fields (link (a) below) and showed this as well. There is a discussion about a negatively charged surface in base creating a more rigid water network as it interacts strongly with the electric field and is therefore more difficult to reorganize during the charge transfer through the double layer as shown by Marc Koper (link (b) below). I would really like the authors to comment on this and maybe include this in the manuscript as it is highly relevant. It could be that MgO acts just like Ni(OH)₂ clusters to improve activity rather than having any other effect.

a) <https://pubs.acs.org/doi/abs/10.1021/jacs.9b05148> (Tracking Electrical Fields at the Pt/H₂O Interface during Hydrogen Catalysis)

b) <https://www.nature.com/articles/nenergy201731> (Interfacial water reorganization as a pH-dependent descriptor of the hydrogen evolution rate on platinum electrodes)

Reply: We thank the reviewer for the thought-provoking comment and kind recommendation to these excellent articles. As the reviewer commented, Jaeyune Ryu *et al.* and Koper *et al.* proposed that strong electric field exists at the electrode/electrolyte interface. The strength of the interfacial electric field, as obtained by the difference between the potential of zero free charge (pzfc) and the actual applied electrode potential, seriously affects the electrochemical reactivity. In alkaline medium, the pzfc of Pt (111) is far from the hydrogen region and close to the OH_{ads} region, leading to a large reorganization energy for interfacial water when OH⁻ transfers through the interfacial double layer, and resulting in a higher energetic barrier of the Volmer step.

In our work, different from the shuffle OH⁻ throughout the double layer that is strongly dependent on the reorganization of interfacial water [Nature Energy 2017, 2, 17031], the energy of reorganization of the interfacial water to move the H⁺ through the double layer is relatively small. Xiao Liang Hu *et al.* show that adsorbed water clusters of just two molecules are sufficient to facilitate small proton transfer between oxygen atoms [Phys. Chem. Chem. Phys., 2010, 12, 3953–3956]. The proton is translocated after a series of successive protonation-dissociation steps where hydrogen ions hop from oxygen to oxygen. **More importantly, an additional driving force for H⁺ accumulation exists in the Pt^{δ-}/MgO system, that is, the electrostatic attraction between the positively-charged H₃O⁺ and the negatively-charged Pt^{δ-}.** Thus the Pt^{δ-} accelerates the H₃O⁺ migration and an acid-like environment is formed around the Pt^{δ-} in alkaline medium. Besides, the presence of H₃O⁺ negatively shifts the pzfc so as to improve the HER kinetics. This has been verified by our calculations. As shown in Fig. 5d, 5e, the migration energy of H⁺ decreases gradually as the H₃O⁺ migrates closer to Pt^{δ-}, indicating the strong tendency of accumulating H⁺ around the Pt^{δ-} nanoparticles.

Lastly, about the role of Ni(OOH)₂ in alkaline HER. First of all, Marković *et al.* reported that the Ni(OOH)₂ could promote the dissociation of water, the adsorption of hydrogen on the Pt surfaces and recombination into molecular hydrogen [Science 2011, 334, 1256]. Besides, Koper *et al.* suggested that the Ni(OOH)₂ shifts the pzfc toward the HER equilibrium potential, then promotes the hydrogen evolution reaction by

lowering the energy barrier necessary for the reorganization of the interfacial water network. The only function of MgO similar to that of Ni(OH)₂ is to promote the water dissociation. However, for Ni(OH)₂, the amount of protons produced at the interface is limited and free, which cannot solve the fundamental problem of much lower kinetics of alkaline HER than acidic HER. Our oxygen vacancies rich MgO can not only produce a large number of protons, **but also create negatively-charged Pt^{δ-}**. Moreover, negative valence Pt^{δ-} is conducive to the accumulation of H⁺, which modifies the local reaction environment around Pt.

Revision:

Accordingly, in the revised manuscript, we have added the statements: *“Then the Pt^{δ-} accelerates H₃O⁺ migration to form a local acid-like environment around Pt^{δ-} in alkaline medium.” have been changed to “Due to the lower migration energy for the H⁺ and the additional electrostatic attraction between the positively-charged H₃O⁺ and the negatively-charged Pt^{δ-}, the H⁺ is easy to migrate throughout the rigid water network of the double layer which caused by the positive shift of potential of zero free charge (pzfc) to form a local acid-like environment around Pt^{δ-}.”*

5) The XPS and XANES for Pt/C in fig 3d and 3e might show that the surface is positively charged but under applied potential at HER relevant potentials this is highly unlikely, and I would think that the surface would be negatively charged at those potentials considering the information we have on the pzfc of the Pt surface. Can the authors comment on how they think this result is relevant at HER relevant potentials?

Reply: We totally agree with the reviewer that Pt/C may be negatively charged if an appropriate HER relevant potential is applied. However, only the negatively-charged Pt induced by external electric field cannot form a local acidic environment, so it cannot fundamentally solve the slow reaction kinetics of poor alkaline HER performance. In the alkaline medium, pzfc shifts positively, which is far from the hydrogen region. As a result, the HER potential is more negative to the pzfc, resulting in the higher energetic barrier for the HER. However, for the Pt/MgO, the multiple physicochemical interactions between the V_O-rich MgO, negatively charged Pt^{δ-} and reaction intermediate create a beneficial local acid-like environment for the Pt. If we consider these interactions within the framework of the pzfc theory, the local acidic environment shifts the pzfc toward the HER equilibrium potential, and then promotes the hydrogen evolution reaction by lowering the energy barrier.

There are two important requirements for the formation of local acidic environment: sufficient water dissociation ability and proton aggregation ability. One good example is that Qiao's group observed a local H_3O^+ reaction environment at a specific pH condition (0.1–1M KOH, pH=13–13.5) during the HER process [Nat. Commun. 2019, 10, 4876, reference 24]. Further analysis shows that the complex surface structure of nanoparticles allows for the existence of a variety of different active sites to facilitate water dissociation and to concentrate the H^+ groups. Since these two key processes can proceed at the same time, an abundance of H_3O^+ is accumulated. However, the active site is not identified and the phenomenon can only be observed under this specific condition. Besides, using the Gouy–Chapman–Stern theory, Adeela Nairan *et. al* proposed that the locally enhanced electrostatic field induced by the high curvature surfaces facilitated fast proton generation and accumulation, thus forming a “pseudo acidic” local environment [Energy Environ. Sci., 2021,14, 1594-1601]. It is our aim in our work to seek a suitable system capable of creating and engineering the desired local acidity under alkaline conditions, as well as to elucidate the underlying mechanism.

Revision:

Accordingly, in the revised manuscript, we have added the statements: “*Within the framework of the potential of zero free charge (pzfc), the local acid-like environment will shift the pzfc of Pt toward the HER equilibrium potential, and then promotes the hydrogen evolution reaction by lowering the energy barrier.*”

6) Gibbs free energy of hydrogen adsorption ($\Delta\text{G}_{\text{H}^*}$) is a reasonable descriptor in acid but in base it gets more complicated which is why there is a lot of discussion on trying to identify an accurate descriptor in base. While it is still important to optimize hydrogen adsorption, most consider it an inaccurate descriptor in base just because it cannot explain everything. This study lacks any literature study about the different mechanisms in base which may be important to provide context to this study.

Reply: Thanks for the reviewer's constructive comment. As the reviewer commented, Gibbs free energy of hydrogen adsorption ($\Delta\text{G}_{\text{H}^*}$) is a universal descriptor in acid. But in base it's still in dispute, and there are two main different viewpoints of explaining this. Yan *et al.* demonstrated that a pH dependent H-binding energy (HBE) lies at the origin of the pH dependent kinetics, and proposed that the HBE was the sole descriptor for the HER/HOR [Sci. Adv. 2016, 2, e1501602., Nat. Commun. 2015, 6, 5848, Energy & Environmental Science 2013, 6, 1509-1512.]. Meanwhile, Koper *et al.* also proposed

that the H_{upd} is unlikely to be associated with the adsorption of hydrogen alone, and the oxygenated species adsorption needs to be considered as well [Catal. Today 2013, 202, 105-113; Phys. Chem. Chem. Phys. 2010, 12, 15217-15224]. In addition, Marković et al. argued that adsorbed hydroxyl (OH^*) may affect the kinetics of the HER by competing for one single active site with H^* [Nat. Mater. 2012, 11, 550; Science 2011, 334, 1256]. Despite these proposed mechanisms, the key descriptor governing alkaline HER activities is still under debate.

In this work, our use of the Gibbs free energy of hydrogen adsorption (ΔG_{H^*}) as a descriptor of hydrogen adsorption in the alkaline medium is rationalized by the fact that the active site (Pt) related to the step of hydrogen adsorption was surrounded by the H_3O^+ abundant, acid-like environment. This also agrees with the kinetic analysis giving a Tafel slope of 39 mV/dec in alkaline solution, which indicates that the H_2 recombination is the rate determining step for the overall HER in Pt/MgO. The reason is that MgO is beneficial for facilitating the dissociation of H_2O molecule, giving rise to abundant H_3O^+ groups, and the generated H_3O^+ groups migrate to negatively charged Pt^{δ^-} and accumulates around Pt^{δ^-} nanoparticles. As a result, the Pt/MgO catalyst creates a local acid-like environment in alkaline solution, which provides Pt with a favorable reaction environment for HER in alkaline condition.

Revision,

Accordingly, in the revised manuscript, the original statements “*Considering the experimental results that V_O -rich MgO creates a local acid-like environment for Pt, we then examined how the different charge state (Pt^{δ^-} , Pt^0 and Pt^{δ^+}) of the Pt nanoparticles could influence the activity of HER (Supplementary Fig.22)*” have been changed to “*Considering the fact that V_O -rich MgO creates a local acid-like environment for negatively charged Pt^{δ^-} , we calculated the Gibbs free energy of hydrogen adsorption (ΔG_{H^*}), which is widely accepted as a universal descriptor for HER in acid, to judge the activity of Pt in the step of hydrogen adsorption. And we examined the effects of three different charge state (Pt^{δ^-} , Pt^0 and Pt^{δ^+}) of the Pt nanoparticles on the activity of HER (Supplementary Fig. 40)*”

7) In supplementary Fig. 23, can the authors comment on the reproducibility of the CV in 1 M KOH. Generally, in my experience I have seen (110) and (100) peaks appear quite clearly in the CVs in acid and base. Are the CVs shown after a steady state is reached and there is no more change? Could the authors comment on the reproducibility

of the results including the CVs? It would also be useful to know what hardware is used for KOH results. Whether it is glass or FEP based? How is it cleaned before tests?

Reply: We really thank the reviewer for reminding us these important points. According to your suggestion, we retested the CVs. Based on the guideline of Noémie Elgrishi *et. al* [J. Chem. Educ. 2018, 95, 2, 197-206], first, we bubbled inert gas (nitrogen) through the solution to remove oxygen. In order to minimise the risk of any contamination by water, we heated the components in a glassware oven prior to use. Then we retested the CVs of Pt/C and Pt/MgO. Figure R28a displays the first 50 cycles of CV curves for Pt/MgO in 1M KOH., It can be seen that the curve is stable at the second cycle, and the (110) and (100) peaks appear quite clearly in the CVs. From Figure R28b it is obvious that the under-potential deposition H (H_{upd}) peak of Pt/MgO still shifts to a lower potential in comparison to Pt/C, manifesting the weaker H^* adsorption on MgO, consistent with the results of DFT calculations.

Figure R28: (a) The first 50 cycles of CV curves for Pt/MgO in 1M KOH. (b) The steady CV curves of Pt/MgO and Pt/C.

Revision,

In the revised manuscript, we used the new Cyclic voltammetry (CV) curves to replace the previous one.

REVIEWERS' COMMENTS

Reviewer #1 (Remarks to the Author):

The new results clearly support the conclusions.

Reviewer #2 (Remarks to the Author):

This is the revision of the previously submitted manuscript. The authors have made significant efforts to improve the quality of the manuscript and have addressed most of my comments. Therefore, I suggest this manuscript for publication.

Reviewer #3 (Remarks to the Author):

The authors have provided a detailed response to my questions which I really appreciate. Several modifications have been made and the errors I had pointed out have been corrected. Although there are certain areas where I have my disagreements, I would like to acknowledge that many conclusions made are supported by the advanced operando techniques. This study also provides a methodology on how to create an acid like local environment which can affect the kinetics of the reaction. I recommend this manuscript to be published in Nature Communications but would advise to proof read the manuscript to eliminate any grammatical mistakes.

Response to Reviewers

Reviewer #1 (Remarks to the Author):

The new results clearly support the conclusions.

Reply: We sincerely thank the reviews for spending the valuable time on reviewing our manuscript. All the insightful comments and suggestions from the reviewer are very helpful for making the manuscript higher quality.

Reviewer #2 (Remarks to the Author):

This is the revision of the previously submitted manuscript. The authors have made significant efforts to improve the quality of the manuscript and have addressed most of my comments. Therefore, I suggest this manuscript for publication.

Reply: We truly appreciate the reviewer's suggestion of our manuscript for publication. We are glad that our revised version of the manuscript satisfies the reviewer. We would like to thank again the reviewer's constructive guidance and suggestions for greatly improving the quality of the manuscript.

Reviewer #3 (Remarks to the Author):

The authors have provided a detailed response to my questions which I really appreciate. Several modifications have been made and the errors I had pointed out have been corrected. Although there are certain areas where I have my disagreements, I would like to acknowledge that many conclusions made are supported by the advanced operando techniques. This study also provides a methodology on how to create an acid like local environment which can affect the kinetics of the reaction. I recommend this manuscript to be published in Nature Communications but would advise to proof read the manuscript to eliminate any grammatical mistakes.

Reply: We appreciate the reviewer's acknowledgement that our operando measurements support many of our conclusions in this work, as well as the recommendation of this manuscript to be published in Nature Communications. We are grateful to the reviewer's high evaluation of the scientific merit of this work that it provides a methodology on how to create an acid like local environment which can affect the kinetics of the reaction. We would also like to thank the reviewer for his/her thoughtful comments and suggestions on improving our research work. In addition, as suggested, we have used the Nature Research Editing Service to edit the English of this manuscript.